# CHAIN OF LOG-CONCAVE MARKOV CHAINS

**Saeed Saremi[1], Ji Won Park[1], Francis Bach[2]**
[1]Frontier Research, Prescient Design, Genentech, South San Francisco, CA
[2]Inria, Ecole Normale Supérieure, Université PSL, Paris, France

## ABSTRACT

We introduce a theoretical framework for sampling from unnormalized densities based on a smoothing scheme that uses an isotropic Gaussian kernel with a single fixed noise scale. We prove one can decompose sampling from a density (minimal assumptions made on the density) into a sequence of sampling from log-concave conditional densities via accumulation of noisy measurements with equal noise levels. Our construction is unique in that it keeps track of a history of samples, making it non-Markovian as a whole, but it is lightweight algorithmically as the history only shows up in the form of a running empirical mean of samples. Our sampling algorithm generalizes walk-jump sampling (Saremi & Hyvärinen, 2019). The "walk" phase becomes a (non-Markovian) chain of (log-concave) Markov chains. The "jump" from the accumulated measurements is obtained by empirical Bayes. We study our sampling algorithm quantitatively using the 2-Wasserstein metric and compare it with various Langevin MCMC algorithms. We also report a remarkable capacity of our algorithm to "tunnel" between modes of a distribution.

## 1 INTRODUCTION

Markov chain Monte Carlo (MCMC) is an important class of general-purpose algorithms for sampling from an unnormalized probability density of the form $p(x) = e^{-f(x)}/Z$ in $\mathbb{R}^d$. This is a fundamental problem and appears in a variety of fields, e.g., statistical physics going back to 1953 (Metropolis et al., 1953), Bayesian inference (Neal, 1995), and molecular dynamics simulations (Leimkuhler & Matthews, 2015). The biggest challenge facing MCMC is that the distributions of interest lie in very high dimensions and are far from being log-concave, therefore the probability mass is concentrated in small pockets separated by vast empty spaces. These large regions with small probability mass make navigating the space using Markov chains very slow. The second important challenge facing MCMC is that the log-concave pockets themselves are typically ill-conditioned—highly elongated, spanning different directions for different pockets—which only adds to the complexity of sampling.

The framework we develop in this paper aims at addressing these problems. The general philosophy here is that of **smoothing**, by which we expand the space from $\mathbb{R}^d$ to $\mathbb{R}^{md}$ for some integer $m$ and "fill up" the empty space iteratively with probability mass in an approximately isotropic manner, the degree of which we can control using a single smoothing (noise) hyperparameter $\sigma$. The map from (noisy samples in) $\mathbb{R}^{md}$ back to (clean samples in) $\mathbb{R}^d$ is based on the **empirical Bayes** formalism (Robbins, 1956; Miyasawa, 1961; Saremi & Hyvärinen, 2019; Saremi & Srivastava, 2022). In essence, a single "jump" using the empirical Bayes estimator removes the masses that were created during sampling. We prove a general result that, for any large $m$, the problem of sampling in $\mathbb{R}^d$ can be reduced to sampling from a sequence of log-concave densities: **once log-concave, always log-concave.** The trade-off here is the linear time cost of accumulating noisy measurements over $m$ iterations.

More formally, instead of sampling from $p(x)$, we sample from the density $p(y_{1:m})$ that is associated with $Y_{1:m} := (Y_1, \ldots, Y_m)$, where $Y_t = X + N_t$, $t \in [m]$, $N_t \perp X$, and $N_t \sim \mathcal{N}(0, \sigma^2 I)$ all independent. As we show in the paper, there is a duality between sampling from $p(x)$ and sampling from $p(y_{1:m})$ in the regime where $m^{-1/2}\sigma$ is small, irrespective of how large $\sigma$ is. This is related to the notion of **universality** class underlying the smoothed densities. Crucial to our formalism is keeping track of the history of all the noisy samples generated along the way using the factorization

$$p(y_{1:m}) = p(y_1) \prod_{t=2}^{m} p(y_t|y_{1:t-1}). \tag{1.1}$$

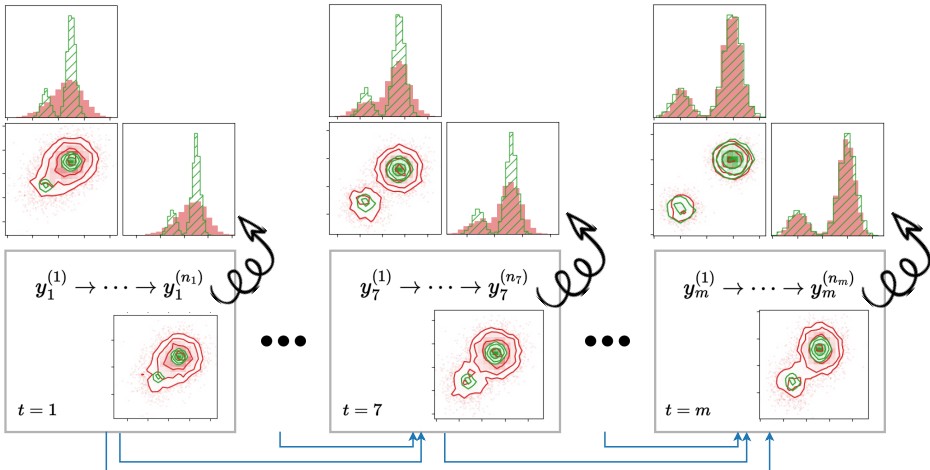

Figure 1: Chain of log-concave Markov chains. Here, $(y_t^{(i)})_{i \in [n_t]}$ are samples from a Markov chain, which is used to generate independent draws from $p(y_t|y_{1:t-1})$ for $t \in [m]$. The blue arrows indicate the non-Markov aspect of our sampling scheme: the accumulation of noisy measurements. The wiggly arrows indicate the denoising "jumps". In this example, $p(y_t|y_{1:t-1})$ is log-concave for all $t$, but the jumps asymptotically sample the target density (a mixture of two Gaussians) as $t$ increases.

An important element of this sampling scheme is therefore **non-Markovian**. However, related to our universality results, this history only needs to be tracked in the form of an empirical mean, so the memory footprint is minimal from an algorithmic perspective. See Fig. 1 for a schematic.

A more technical summary of our contributions and the outline of the paper are as follows:

- In Sec. 2, we prove **universality** results underlying the smoothed densities $p(y_{1:m})$.

- We study anisotropic Gaussians in Sec. 3, proving a *negative result* regarding the condition number of $p(y_{1:m})$ in comparison to $p(y_1)$ in the same universality class. This analysis becomes a segue to our factorization (1.1), where in remarkable contrast we show that the condition number **monotonically improves** upon accumulation of measurements.

- Sec. 4 is at the heart of the paper, where we prove several results culminating in Theorem 1, which shows that a broad class of sampling problems can be transformed into a sequence of sampling from strongly **log-concave** distributions using our measurement accumulation scheme. (This is a feasibility result; in particular, we do not prove here that the log-concave sampling strategy is optimal.) We examine the theorem by algebraically studying an example of a mixture of Gaussians in detail. In Sec. 4, we also outline our general sampling algorithm.

- We validate our algorithm on carefully designed test densities in Sec. 5. In particular, our algorithm results in lower 2-Wasserstein metric compared to sampling from $p(x)$ using Langevin MCMC (without any smoothing). We also qualitatively report the capacity of our log-concave sampling scheme to **tunnel** to a mode of a distribution with a small probability mass in a small number of steps when it is initialized at a mode with much higher mass.

## 1.1 RELATED WORK

Our solution, sketched above, has its roots in **walk-jump** sampling (Saremi & Hyvärinen, 2019) and its recent generalization (Saremi & Srivastava, 2022). Both papers were framed within the context of generative modeling, i.e., sampling from an *unknown* distribution from which one has access to independent samples. In contrast, this work lays the theoretical foundation for the fundamental problem of sampling from an unnormalized density when there are no samples available. In addition, regarding the recent development, we show analytically that the intuition expressed by Saremi & Srivastava (2022) regarding the distribution $p(y_{1:m})$ being well-conditioned is not correct. This nontrivial negative result motivates our analysis of the non-Markovian scheme for sampling $p(y_{1:m})$.

Our methodology is agnostic to the algorithm used for sampling from $p(y_t|y_{1:t-1})$ in (1.1). However, we have been particularly motivated by the research on **Langevin MCMC** which is a class of gradient-based sampling algorithms obtained by discretizing the Langevin diffusion (Parisi, 1981). There is a growing body of work on the analysis of Langevin MCMC algorithms of various complexity (overdampled, Metropolis-adjusted, underdamped, higher-order) for sampling from log-concave distributions (Dalalyan, 2017; Durmus & Moulines, 2017; Cheng et al., 2018; Dwivedi et al., 2018; Shen & Lee, 2019; Cao et al., 2021; Mou et al., 2021; Li et al., 2022).

There is a significant body of work on **sequential** methods for sampling, rooted in annealing methods in optimization (Kirkpatrick et al., 1983), which became popular in the MCMC literature due to Neal's seminal paper on annealed importance sampling (Neal, 2001). Diffusion models (Sohl-Dickstein et al., 2015; Ho et al., 2020) are a related class of sequential methods for generative modeling. Our sequential scheme is distinguished from earlier methods from separate angles: **(I)** Although we sample the conditional densities with Markov chains, we condition on all the previous samples that were generated. As a whole, our scheme is strictly *non-Markovian*. **(II)** In our sequential scheme, we are able to guarantee that we sample from (progressively more) *log-concave* densities. To our knowledge, no other sampling frameworks can make such guarantees. **(III)** Compared to diffusion models, the noise level in our framework is held *fixed*. This is an important feature of our sampling algorithm and it underlies many of its theoretical properties. **(IV)** All prior sequential schemes rely on a noising/annealing *schedule* which is hard to tune, and their performance is sensitive to the choice of the schedule (Karras et al., 2022; Syed et al., 2022). In contrast, our sequential scheme is free of scheduling and relies on only two parameters: the noise level $\sigma$ and the number of measurements $m$.[1]

**Notation.** We use $p$ to denote probability density functions and adopt the convention where we drop the random variable subscript to $p$ when the arguments are present, e.g., $p(x) \coloneqq p_X(x)$, $p(y_2|y_1) \coloneqq p_{Y_2|Y_1=y_1}(y_2)$. We reserve $f$ to be the energy function associated with $p(x) \propto e^{-f(x)}$. We use $\lambda$ to denote the spectrum of a matrix, e.g., $\lambda_{\max}(C)$ is the largest eigenvalue of $C$. We use the shorthand notations $[m] = \{1, \ldots, m\}$, $y_{1:m} = (y_1, \ldots, y_m)$, and $\overline{y}_{1:m} = \frac{1}{m}\sum_{t=1}^{m} y_t$.

## 2 UNIVERSAL $(\sigma, m)$-DENSITIES

Consider the multimeasurement (factorial kernel) generalization of the kernel density by Saremi & Srivastava (2022) for $m$ isotropic Gaussian kernels with equal noise level (kernel bandwidth) $\sigma$:

$$p(y_{1:m}) \propto \int_{\mathbb{R}^d} e^{-f(x)} \exp\Big(-\frac{1}{2\sigma^2}\sum_{t=1}^{m}\|x - y_t\|^2\Big)dx. \tag{2.1}$$

We refer to $p(y_{1:m})$ as the $(\sigma, m)$-*density*. Equivalently, $Y_t|x \overset{\text{iid}}{\sim} \mathcal{N}(x, \sigma^2 I)$, $t \in [m]$. Clearly, $p(y_{1:m})$ is permutation invariant $p(y_1, \ldots, y_m) = p(y_{\pi(1)}, \ldots, y_{\pi(m)})$, where $\pi : [m] \to [m]$ is a permutation of the $m$ measurements. We set the stage for the remainder of the paper with a calculation that shows the permutation invariance takes the following form (see Appendix A):

$$\log p(y_{1:m}) = \varphi(\overline{y}_{1:m}; m^{-1/2}\sigma) + \frac{m}{2\sigma^2}\Big(\|\overline{y}_{1:m}\|^2 - \frac{1}{m}\sum_{t=1}^{m}\|y_t\|^2\Big) + \mathrm{cst}, \tag{2.2}$$

where

$$\varphi(y; \sigma) \coloneqq \log \int e^{-f(x)} \exp\Big(-\frac{1}{2\sigma^2}\|x - y\|^2\Big)dx. \tag{2.3}$$

The calculation is straightforward by grouping the sums of squares in (2.1):

$$-\sum_{t=1}^{m}\|x - y_t\|^2 = m\Big(-\|x - \overline{y}_{1:m}\|^2 + \|\overline{y}_{1:m}\|^2 - \frac{1}{m}\sum_{t=1}^{m}\|y_t\|^2\Big) + \mathrm{cst}.$$

In addition, the Bayes estimator of $X$ given $Y_{1:m} = y_{1:m}$ simplifies as follows (see Appendix A):

$$\mathbb{E}[X|y_{1:m}] = \overline{y}_{1:m} + m^{-1}\sigma^2 \nabla\varphi(\overline{y}_{1:m}; m^{-1/2}\sigma). \tag{2.4}$$

These calculations bring out a notion of universality class that is associated with $p(y_{1:m})$ formalized by the following definition and proposition.

---

[1]Our method can be viewed as a discretization scheme in the *stochastic localization* method, which we plan to formalize in future research.

**Definition 1** (Universality Class). *We define the universality class $[\tilde{\sigma}]$ as the set of densities $p(y_{1:m})$, in the family of $(\sigma, m)$-densities, such that for all $(\sigma, m) \in [\tilde{\sigma}]$ the following holds: $m^{-1/2}\sigma = \tilde{\sigma}$.*

**Proposition 1.** *If $Y_{1:m} \sim p(y_{1:m})$, let $\hat{p}_{\sigma,m}$ be the distribution of $\mathbb{E}[X|Y_{1:m}]$, and define $\hat{p}_\sigma = \hat{p}_{\sigma,1}$. Then $\hat{p}_{\sigma,m} = \hat{p}_{m^{-1/2}\sigma}$. In other words, $\hat{p}_{\sigma,m}$ is identical for densities in the same universality class.*

*Proof.* We are given $X \sim e^{-f(x)}$, $Y_t = X + \varepsilon_t$, $\varepsilon_t \sim \mathcal{N}(0, \sigma^2 I)$ independently for $t \in [m]$. It follows $\overline{Y}_{1:m} = X + \tilde{\varepsilon}$, where $\tilde{\varepsilon} \sim \mathcal{N}(0, \tilde{\sigma}^2 I)$, where $\tilde{\sigma}^2 = m^{-1}\sigma^2$. Using (2.4), $\mathbb{E}[X|y_{1:m}]$ is distributed as

$$X + \tilde{\varepsilon} + \tilde{\sigma}^2 \nabla\varphi(X + \tilde{\varepsilon}; \tilde{\sigma}),$$

which is identical for all densities $p(y_{1:m})$ in $[\tilde{\sigma}]$. □

### 2.1 DISTRIBUTION OF $\mathbb{E}[X|y_{1:m}]$ VS. $p_X$: UPPER BOUND ON THE 2-WASSERSTEIN DISTANCE

Our goal is to obtain samples from $p_X$, but in walk-jump sampling the samples are given by $\mathbb{E}[X|y_{1:m}]$, where $y_{1:m} \sim p(y_{1:m})$ (Saremi & Hyvärinen, 2019; Saremi & Srivastava, 2022). Next, we address how far $\hat{p}_{\sigma,m}$ is from the density of interest $p_X$.

**Proposition 2.** *The squared 2-Wasserstein distance between $p_X$ and $\hat{p}_{\sigma,m}$ is bounded by*

$$W_2(p_X, \hat{p}_{\sigma,m})^2 \leqslant \frac{\sigma^2}{m} d.$$

The proof is given in Appendix B. As expected, the upper bound is expressed in terms of $\tilde{\sigma}^2 = \sigma^2/m$. A close inspection of the proof shows that the bound above is loose as it is obtained from the rate resulting from replacing the empirical Bayes estimator $\mathbb{E}[X|y_{1:m}]$ with the empirical mean $\overline{Y}_{1:m}$. Note, however, that when the prior $p(x)$ is "strong" (e.g., low entropy), the dependence on $\sigma^2/m$ can be significantly improved.

## 3 THE GEOMETRY OF $(\sigma, m)$-DENSITIES

In this section we analyze at the problem of sampling from $p(y_{1:m})$ where we consider $p(x)$ to be an anisotropic Gaussian, $X \sim \mathcal{N}(0, C)$, with a diagonal covariance matrix:

$$C = \text{diag}(\tau_1^2, \ldots, \tau_d^2). \tag{3.1}$$

The density $p_X$ is strongly log-concave with the property $\tau_{\max}^{-2} I \preccurlyeq \nabla^2 f(x) \preccurlyeq \tau_{\min}^{-2} I$, therefore its condition number is $\kappa = \tau_{\min}^{-2} \tau_{\max}^2$. Log-concave densities with $\kappa \gg 1$ are considered ill-conditioned. Since $Y_1 \sim \mathcal{N}(0, C + \sigma^2 I)$, the condition number for (single-measurement) smoothed density, which we denote by $\kappa_{\sigma,1}$ is given by:

$$\kappa_{\sigma,1} = (1 + \sigma^{-2}\tau_{\max}^2)/(1 + \sigma^{-2}\tau_{\min}^2). \tag{3.2}$$

Next, we give the full spectrum of the precision matrix associated with $(\sigma, m)$-densities.

**Proposition 3.** *Consider an anisotropic Gaussian density $X \sim \mathcal{N}(0, C)$ in $\mathbb{R}^d$, where $C_{ij} = \tau_i^2 \delta_{ij}$. Then the $(\sigma, m)$-density is a centered Gaussian in $\mathbb{R}^{md}$: $Y_{1:m} \sim \mathcal{N}(0, F_{\sigma,m}^{-1})$. For $m \geqslant 2$, the precision matrix $F_{\sigma,m}$ is block diagonal with $d$ blocks (indexed by $i$) of size $m \times m$, each with the following spectrum: (i) There are $m - 1$ degenerate eigenvalues equal to $\sigma^{-2}$, (ii) The remaining eigenvalue equals to $(\sigma^2 + m\tau_i^2)^{-1}$. The condition number $\kappa_{\sigma,m}$ associated with the $(\sigma, m)$-density is given by:*

$$\kappa_{\sigma,m} = \frac{\lambda_{\max}(F_{\sigma,m})}{\lambda_{\min}(F_{\sigma,m})} = 1 + m \cdot \sigma^{-2}\tau_{\max}^2.$$

**Remark 1** (The curse of sampling all measurements at once). *The above proposition is a negative result regarding sampling from $p(y_{1:m})$ if—this is an important "if"—all $m$ measurements $y_{1:m}$ are sampled in parallel (at the same time). This is because $m\sigma^{-2} = \tilde{\sigma}^{-2}$ remains constant for $m > 1$ for $(\sigma, m) \in [\tilde{\sigma}]$—even worse, the condition number $\kappa_{\sigma,m}$ is strictly greater than $\kappa_{\tilde{\sigma},1}$ for $m > 1$.*

This negative result regarding the sampling scheme by Saremi & Srivastava (2022), we call *joint multimeasurement sampling* (JMS), leads to our investigation below into sampling from $p(y_{1:m})$ sequentially using the factorization (1.1) that we call *sequential multimeasurement sampling* (SMS). Now, we perform the analysis in Proposition 3 for the spectrum of the conditional densities in (1.1).

**Proposition 4.** *Assume $X \sim \mathcal{N}(0, C)$ is the anisotropic Gaussian in Proposition 3. Given the factorization of $p(y_{1:m})$ in (1.1), for $t > 1$, the conditional density $p(y_t|y_{1:t-1})$ is a Gaussian with a shifted mean, and with a diagonal covariance matrix:*

$$-2\sigma^2 \log p(y_t|y_{1:t-1}) = \sum_{i=1}^{d} (1 - A_{ti}) \cdot \left(y_{ti} - \frac{A_{ti}}{1 - A_{ti}} \sum_{k=1}^{t-1} y_{ki}\right)^2 + \text{cst},$$

*where $A_{ti}$ is short for $A_{ti} = \left(t + \sigma^2 \tau_i^{-2}\right)^{-1}$. The precision matrix associated with $p(y_t|y_{1:t-1})$, denoted by $F_{t|1:t-1}$, has the following spectrum*

$$\sigma^2 \lambda_i(F_{t|1:t-1}) = 1 - \left(t + \sigma^2 \tau_i^{-2}\right)^{-1},$$

*with the following condition number*

$$\kappa_{t|1:t-1} = \frac{1 - (t + \sigma^2 \tau_{\min}^{-2})^{-1}}{1 - (t + \sigma^2 \tau_{\max}^{-2})^{-1}}.$$

*Lastly, the condition number $\kappa_{t|1:t-1}$ is monotonically decreasing as $t$ increases (for any $m > 1$):*

$$1 < \kappa_{m|1:m-1} < \cdots < \kappa_{3|1:2} < \kappa_{2|1} < \kappa_1, \tag{3.3}$$

*where $\kappa_1 := \kappa_{\sigma,1}$ is given by (3.2).*

The proofs for Proposition 3 and Proposition 4 are given in Appendix C. These two propositions stand in a clear contrast to each other: in the SMS setting of Proposition 4, sampling becomes *easier* by increasing $t$ as one goes through accumulating measurements $y_{1:t}$ sequentially, where in addition $\kappa_1$ can itself be decreased by increasing $\sigma$. Next, we analyze the SMS scheme in more general settings.

## 4 CHAIN OF LOG-CONCAVE MARKOV CHAINS

Can we devise a sampling scheme where we are guaranteed to always sample log-concave densities? This section is devoted to several results in that direction. We start with the following two lemmas.

**Lemma 1.** *Assume $\forall x \in \mathbb{R}^d$, $\nabla^2 f(x) \preccurlyeq LI$ and $\|\nabla f(x)\| \geqslant \mu\|x - x_0\| - \Delta$ for some $x_0$. Then, $\forall y \in \mathbb{R}^d$:*

$$\nabla^2 (\log p)(y) \preccurlyeq \left(-1 + \frac{3Ld}{\mu^2\sigma^2} + \frac{3\Delta^2}{\mu^2\sigma^2} + 3\frac{\|x_0 - y\|^2}{\mu^2\sigma^6}\right)\frac{I}{\sigma^2}.$$

The proof is given in Appendix D.

**Lemma 2.** *Consider the density $p(x)$ associated with the random variable $X$ in $\mathbb{R}^d$ and the $(\sigma, m)$-density given by (2.1). Then in expectation, for any $m \geqslant 1$ the conditional densities become more log-concave upon accumulation of measurements:[2]*

$$\mathbb{E}_{y_1} \nabla_{y_1}^2 \log p(y_1) \succcurlyeq \mathbb{E}_{y_{1:2}} \nabla_{y_2}^2 \log p(y_2|y_1) \succcurlyeq \cdots \succcurlyeq \mathbb{E}_{y_{1:m}} \nabla_{y_m}^2 \log p(y_m|y_{1:m-1}).$$

*Proof.* The full proof of the lemma is given in Appendix D, where we derive the following:

$$\nabla_{y_m}^2 \log p(y_m|y_{1:m-1}) = -\sigma^{-2} I + \sigma^{-4} \text{cov}(X|y_{1:m}).$$

The proof follows through since due to the law of total covariance the mean of the posterior covariance $\mathbb{E}_{y_{1:m}} \text{cov}(X|y_{1:m})$ can only go down upon accumulation of measurements. $\square$

These two lemmas paint an intuitive picture that we expand on in the remainder of this section: (i) by increasing $\sigma$ we can transform a density to be strongly log-concave (Lemma 1) which we can sample our first measurement from, (ii) and by accumulation of measurements we expect sampling to become easier, where in Lemma 2 this is formalized by showing that on average the conditional densities become more log-concave by conditioning on previous measurements. Next, we generalize these results with our main theorem, followed by an example on a mixture of Gaussians.

---

[2]Note that here no assumption is made on the smoothness of $p(x)$.

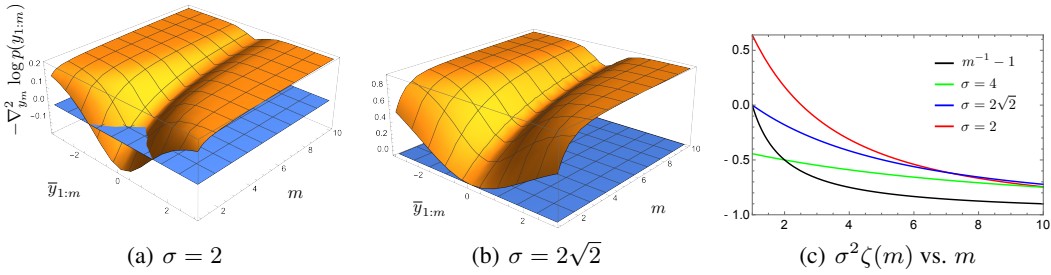

(a) $\sigma = 2$          (b) $\sigma = 2\sqrt{2}$          (c) $\sigma^2 \zeta(m)$ vs. $m$

Figure 2: (a,b) The negative conditional Hessian for two values of $\sigma$ are plotted as a function of $\overline{y}_{1:m}$ and $m$ assuming $X$ is distributed according to (4.4) in 1D, where we set $\mu = 3, \tau = 1$ (see (4.5)). (c) The upper bound in (4.1) is sharp for this example; $\sigma^2 \zeta(m)$ is plotted vs. $m$ for different $\sigma$.

**Theorem 1** (Once log-concave, always log-concave). *Consider $Z$ to be a random variable in $\mathbb{R}^d$ with a compact support, i.e., almost surely $\|Z\|^2 \leqslant R^2$, and take $X = Z + N_0$, $N_0 \perp Z$, $N_0 \sim \mathcal{N}(0, \tau^2 I)$. Then, for any $m \geqslant 1$, the conditional Hessian is upper bounded*

$$\nabla^2_{y_m} \log p(y_m | y_{1:m-1}) \preccurlyeq \zeta(m) I, \tag{4.1}$$

*where:*

$$\zeta(m) = \frac{1}{\sigma^2} \left( \frac{\tau^2}{m\tau^2 + \sigma^2} - 1 \right) + \frac{R^2}{(m\tau^2 + \sigma^2)^2} \tag{4.2}$$

*is a decreasing function of $m$, in particular:*

$$\zeta'(m) = -\frac{\tau^2 (2R^2\sigma^2 + \sigma^2\tau^2 + m\tau^4)}{\sigma^2 (\sigma^2 + m\tau^2)^3} \leqslant 0.$$

*As a corollary, $p(y_1)$ associated with $Y_1 = X + N_1$, $N_1 \sim \mathcal{N}(0, \sigma^2 I)$ is strongly log-concave if*

$$\sigma^2 > R^2 - \tau^2, \tag{4.3}$$

*and stays strongly log-concave upon accumulation of measurements.*

*Proof.* The full proof is given in Appendix D and it is a direct consequence of the following identity:

$$\nabla^2_{y_m} \log p(y_m | y_{1:m-1}) = \frac{1}{\sigma^2} \left( \frac{\tau^2}{m\tau^2 + \sigma^2} - 1 \right) \cdot I + \frac{1}{(m\tau^2 + \sigma^2)^2} \mathrm{cov}(Z | y_{1:m}),$$

which we derive, combined with $\mathrm{cov}(Z | y_{1:m}) \preccurlyeq R^2 I$ due to our compactness assumption. $\qquad \square$

**Remark 2.** *Theorem 1 spans a broad class of sampling problems, especially since $\tau$ can in principle be set to zero. The only property we loose in the setting of $\tau = 0$ is that the upper bound $\zeta(m) I$ does not monotonically go down as measurements are accumulated.*

## 4.1 EXAMPLE: MIXTURE OF TWO GAUSSIANS

In this section we examine Theorem 1 by studying the following mixture of Gaussians for $\alpha = 1/2$:

$$p(x) = \alpha \, \mathcal{N}(x; \mu, \tau^2 I) + (1 - \alpha) \, \mathcal{N}(x; -\mu, \tau^2 I). \tag{4.4}$$

This is an instance of the setup in Theorem 1, where $p(z) = \frac{1}{2}\delta(z - \mu) + \frac{1}{2}\delta(z + \mu)$, and $R^2 = \mu^\top \mu$. By differentiating (2.2) twice we arrive at the following expression for $\nabla^2_{y_m} \log p(y_m | y_{1:m-1})$:

$$\nabla^2_{y_m} \log p(y_m | y_{1:m-1}) = \nabla^2_{y_m} \log p(y_{1:m}) = \sigma^{-2}(m^{-1} - 1)I + m^{-2} H(\overline{y}_{1:m}; m^{-1/2}\sigma), \tag{4.5}$$

where $H(y; \sigma) := \nabla^2 \varphi(y; \sigma)$; see (2.3) for the definition of $\varphi$. In Appendix D we show that for the mixture of Gaussian here, (4.4) with $\alpha = 1/2$, we have

$$H(y; \sigma) = \frac{1}{(\sigma^2 + \tau^2)} \left( -I + \frac{2\mu\mu^\top}{\sigma^2 + \tau^2} \cdot \left( 1 + \cosh\left( \frac{2\mu^\top y}{\sigma^2 + \tau^2} \right) \right)^{-1} \right), \tag{4.6}$$

which takes its maximum at $y = 0$. By using (4.5), it is then straightforward to show that (4.1), (4.2), and (4.3) all hold in this example, with the additional result that the upper bound is now tight. In Fig. 2, these calculations are visualized in 1D for $\mu = 3, \tau = 1$, and for different values of $\sigma$; in panel (c) we also plot $1 - 1/m$ which is the large $m$ behavior of $\sigma^2 \zeta(m)$. This can be seen from two different routes: (4.2) and (4.5).

**Remark 3** (Monotonicity). *The monotonic decrease of the upper bound in Theorem 1, together with the monotonicity result in Lemma 2, may lead one to investigate whether the stronger result*

$$\nabla_{y_1}^2 \log p(y_1) \succcurlyeq \nabla_{y_2}^2 \log p(y_2|y_1) \succcurlyeq \ldots \succcurlyeq \nabla_{y_m}^2 \log p(y_m|y_{1:m-1}), \qquad (4.7)$$

*could hold, e.g., for the mixture of Gaussians we studied here, especially since the upper bound (4.1) is sharp for this example. For (4.7) to hold, $\mathrm{cov}(Z|y_{1:m}) \preccurlyeq \mathrm{cov}(Z|y_{1:m-1})$ almost surely. However, we can imagine a scenario where $y_1 + \cdots + y_{m-1}$ is very large, so that $\mathrm{cov}(Z|y_{1:m}) \approx 0$, while $y_m$ is such that $y_1 + \cdots + y_{m-1} + y_m$ is close to $m\mathbb{E}[Z]$, where $\mathrm{cov}(Z|y_{1:m})$ will be large.*

### 4.2 ALGORITHM: NON-MARKOVIAN CHAIN OF (LOG-CONCAVE) MARKOV CHAINS

Below, we give the pseudo-code for our sampling algorithm. In the inner loop, $\mathrm{MCMC}_\sigma$ is any MCMC method, but our focus in this paper is on Langevin MCMC algorithms[3] that use $\nabla_{y_t} \log p(y_t|y_{1:t-1})$ to sample the new measurement $Y_t$ conditioned on the previously sampled ones $Y_{1:t-1}$.

---

**Algorithm 1:** Sequential multimeasurement walk-jump sampling referred to by SMS. See Fig. 1 for the schematic. A version of $\mathrm{MCMC}_\sigma$ is given in Appendix E.

---

1: **Parameter** noise level $\sigma$
2: **Input** number of measurements $m$, number of steps for each measurement $n_t$
3: **Output** $\hat{X}$
4: Initialize $\overline{Y}_{1:0} = 0$
5: **for** $t = [1, \ldots, m]$ **do**
6:     Initialize $Y_t^{(0)}$
7:     **for** $i = [1, \ldots, n_t]$ **do**
8:         $Y_t^{(i)} = \mathrm{MCMC}_\sigma(Y_t^{(i-1)}, \overline{Y}_{1:t-1})$
9:     **end for**
10:     $Y_t = Y_t^{(n_t)}$
11:     $\overline{Y}_{1:t} = \overline{Y}_{1:t-1} + (Y_t - \overline{Y}_{1:t-1})/t$
12: **end for**
13: **return** $\hat{X} \leftarrow \mathbb{E}[X|Y_{1:m}]$ according to (2.4)

---

#### 4.2.1 ESTIMATING $\nabla \log p(y)$

So far we have assumed we know the smoothed score function $g(y; \sigma) := \nabla(\log p)(y) = \nabla\varphi(y; \sigma)$, and in experiments below we consider cases where we know $g(y; \sigma)$ in closed form. In general settings, we would like to estimate $g$ in terms of the unnormalized $p(x) \propto e^{-f(x)}$. Given (2.3) and (2.4), we write a an expression for $g(y; \sigma)$ which can be turned into various estimators:

$$g(y; \sigma) = \frac{1}{\sigma^2} \cdot \left(\mathbb{E}[\check{X}] - y\right), \ \check{X} \sim e^{-\check{f}(\cdot; y, \sigma)}, \ \text{where } \check{f}(x; y, \sigma) := f(x) + \frac{1}{2\sigma^2}\|x - y\|^2. \quad (4.8)$$

See Appendix F, where we also give two estimators for $g$ depending on how $\mathbb{E}[\check{X}]$ in (4.8) is estimated: $\hat{g}_{\mathrm{plugin}}$ (F.1) is a plug-in estimator obtained by importance sampling, $\hat{g}_{\mathrm{langevin}}$ (F.2) is obtained by sampling $\check{X}$ using Langevin MCMC. Finally, use (2.2) to write $\nabla_{y_t} \log p(y_t|y_{1:t-1})$ in terms of $g$:

$$\nabla_{y_t} \log p(y_t|y_{1:t-1}) = \nabla_{y_t} \log p(y_{1:t}) = \frac{1}{t}g\left(\overline{y}_{1:t}; \frac{\sigma}{\sqrt{t}}\right) + \frac{1}{\sigma^2}\left(\overline{y}_{1:t} - y_t\right). \quad (4.9)$$

The above expression is used in the inner loop of Algorithm 1 to sample from $p(y_t|y_{1:t-1})$ sequentially. We conduct experiments to investigate this aspect of the problem (replacing $g$ with $\hat{g}$) in the appendix.

---

[3]We experimented with a variety of Langevin MCMC algorithms to sample from $p(y_t|y_{1:t-1})$ in the inner loop of Algorithm 1. The results are reported in the appendix due to space constraints. After extensive tuning, we found the algorithm by Sachs et al. (2017) to be the best performing for the test densities we considered.

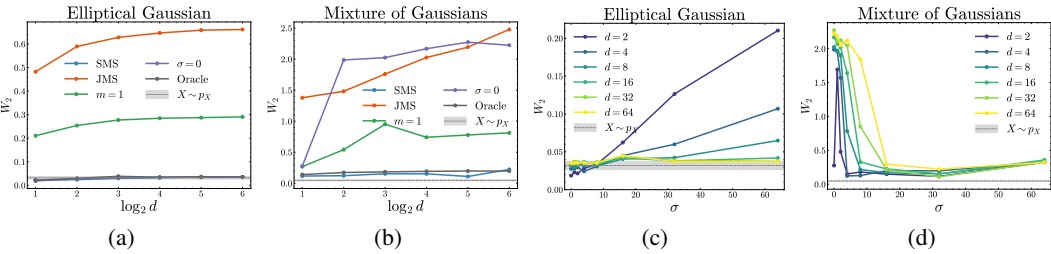

Figure 3: (a, b) Sliced 2-Wasserstein distance vs. $d$ and (c, d) Sliced 2-Wasserstein distance vs. $\sigma$ for varying $d$ for the elliptical Gaussian and Gaussian mixture target densities using SMS.

## 5 EXPERIMENTS

We evaluate the performance of Algorithm 1 alongside related sampling schemes on carefully designed test densities. We compare the following sampling schemes:

- Sequential multimeasurement walk-jump sampling (SMS), Algorithm 1 with $m = 1000$,
- Joint multimeasurement walk-jump sampling (JMS),
- Single-measurement walk-jump sampling ("$m = 1$"),
- Langevin MCMC by Sachs et al. (2017) without any smoothing ("$\sigma = 0$"),
- Exact samples from $\hat{p}_{\sigma,m}$ ("Oracle").[4]

**Metric.** We use the sliced 2-Wasserstein metric (Bonneel et al., 2015; Peyré & Cuturi, 2019) to quantify the consistency of the obtained samples with the target density $p_X$. We use 1,000 projection directions drawn from the Gaussian distribution (Nadjahi et al., 2021, Eq.9).

**MCMC algorithms.** For all the results in this section, we implement MCMC sampling based on underdamped Langevin diffusion (ULD). The particular algorithm used for the results shown in this section extends the BAOAB integration scheme using multiple time steps for the O-part (Sachs et al., 2017). In Appendix H, we present the full comparison across other MCMC algorithms, including other recent ULD variants (Cheng et al., 2018; Shen & Lee, 2019) as well as the Metropolis-adjusted Langevin algorithm (MALA) (Roberts & Tweedie, 1996; Dwivedi et al., 2018).

**Score estimation.** In Appendix I, we compare sampling with the analytic score function, the plug-in estimator of the score function given in (F.1) with varying numbers of MC samples $n$, and the Langevin estimator of the score function given in (F.2).

**Hyperparameter search.** The hyperparameters were tuned for each sampling scheme and the total number of iterations was kept fixed in our comparisons reported here. See Appendix G for details.

### 5.1 ELLIPTICAL GAUSSIAN

The elliptical Gaussian features a poorly conditioned covariance: $X \sim \mathcal{N}(0, C)$, where we set $\tau_1^2 = 0.1, \tau_2^2 = \cdots = \tau_d^2 = 1$ in (3.1). For each $d$, the noise level $\sigma$ and other hyperparameters of the sampling algorithm, such as step size and friction, were tuned. Fig. 3(a) plots the 2-Wasserstein distance with varying $d$. Our main observation here is that SMS outperforms JMS, which is expected from our theoretical analysis in Sec. 3. In addition, as Fig. 3(c) shows, SMS was robust to the choice of $\sigma$ particularly for large $d$ in our experiments. Finally, the underdamped Langevin MCMC by Sachs et al. (2017) does quite well when friction is carefully tuned; we would like to investigate this for larger condition numbers in future research.

---

[4]Oracle is short for the SMS oracle. For the test densities considered here we can draw exact samples from $\hat{p}_{\sigma,m}$, the distribution of $\mathbb{E}[X|Y_{1:m}]$. This baseline is used to separate the issue of the closeness of $\hat{p}_{\sigma,m}$ to $p_X$ from the problem of sampling $\hat{p}_{\sigma,m}$ itself.

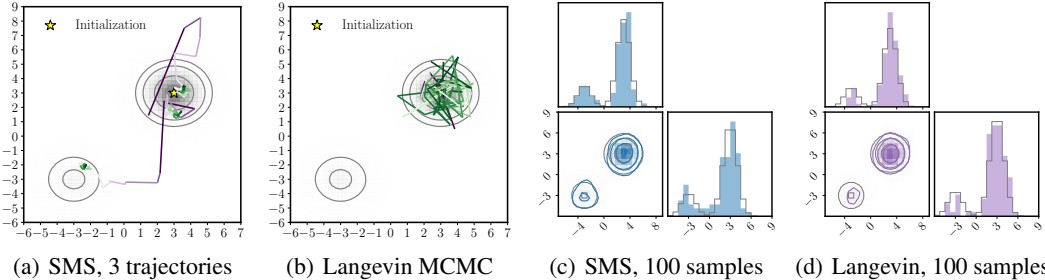

| (a) SMS, 3 trajectories | (b) Langevin MCMC | (c) SMS, 100 samples | (d) Langevin, 100 samples |

Figure 4: Tunneling phenomenon. Trajectories of three walkers (a) under our SMS sampling scheme using Langevin MCMC by Sachs et al. (2017) in the inner loop of Algorithm 1 (b) the same Langevin MCMC without any smoothing. Purple and green indicate the beginning and end of trajectories, respectively. (c, d) The final samples for 100 random trajectories (using identical seeds). Fewer samples reach the smaller mode in Langevin MCMC compared to SMS.

## 5.2 MIXTURE OF GAUSSIANS

To evaluate mixing of multiple modes, we consider the test density (4.4) with $\alpha = 1/5$, $\tau = 1$, and $\mu = 3 \cdot 1_d$, where $1_d$ is the $d$-dimensional vector $(1, \ldots, 1)^\top$. As Fig. 3(a) shows, SMS achieves consistently low (sliced) 2-Wasserstein distance with increasing $d$, whereas other sampling schemes deteriorate in performance. We observe, in Fig. 3(b), that SMS outperforms the best-performing underdamped Langevin MCMC ($\sigma = 0$) in our experiments for at least one $\sigma$ value for all $d$. Higher $d$ requires larger $\sigma$. In addition, we would like to highlight the following: **(I)** Vanilla walk-jump sampling ($m = 1$) is highly ineffective as the dimensions increase. This is in contrast to the sampling from anisotropic Gaussian (already log-concave) in Sec. 5.1. **(II)** The optimal $\sigma$ here is in fact larger than the noise level needed to make $p(y_1)$ log-concave. This is related to the benefits of sampling from better-conditioned log-concave distributions, which is well-known in the literature. In Appendix J, we include results for a mixture of correlated Gaussians supporting the same conclusion.

## 5.3 TUNNELING PHENOMENA

Fig. 4 illustrates the trajectories of three walkers (a) under our SMS sampling scheme and (b) using Langevin MCMC. Each walker has the same random seed between (a) and (b) and was initialized at $(3, 3)$, the dominant mode with 80% of the mass. With SMS, a walker is able to tunnel to the smaller mode fairly quickly, whereas for Langevin MCMC (without smoothing) all three walkers are stuck around the dominant mode. In panels (c) and (d) we also show the histogram of final samples in the same setup with initialization at $(3, 3)$ for 100 walkers after 100 K steps.

In summary, we have consistently observed that the same (Langevin) MCMC algorithm, when used in the inner loop of Algorithm 1, is more effective than when used without Gaussian smoothing.

## 6 CONCLUSION

In this paper, we established a theoretical framework that reduces the general problem of sampling from an unnormalized distribution to that of log-concave sampling defined by a single noise parameter. We conclude with two main limitations of this work at the present time: **(I)** Our results do not make it clear if the log-concave sampling strategy is optimal. The issue of "optimality" is challenging as it is inherently problem-dependent and additionally depends on the MCMC algorithm used in the inner loop of Algorithm 1. **(II)** Related to the issue of optimality is the fact that for general sampling problems, the smoothed score functions required to sample from $p(y_t|y_{1:t-1})$, $t \in [m]$ using gradient-based methods need to be estimated. This is a complex problem and should be investigated in future research. Finally, an immediate application of the machinery we developed here is the problem of generative modeling, as the $m$ smoothed score functions needed in running Algorithm 1 can indeed be learned (approximated) using empirical Bayes least-squares denoising objectives. This approach is similar to training diffusion models; however, our sampling scheme is fundamentally different, as it relies on the accumulation of measurements, controlled by a single noise parameter, $\sigma$.

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

# A DERIVATIONS FOR SEC. 2

## A.1 DERIVATION FOR (2.2)

In this section we derive a general expression for the $(\sigma, m)$-densities $p(y_{1:m})$, short for $p_\sigma(y_{1:m})$:

$$
\begin{aligned}
p(y_{1:m}) &= \int_{\mathbb{R}^d} p(x) \left( \prod_{t=1}^m p(y_t|x) \right) dx \\
&= \int_{\mathbb{R}^d} \frac{1}{Z} e^{-f(x)} \left( \prod_{t=1}^m \frac{1}{(2\pi\sigma^2)^{d/2}} \exp\left( -\frac{1}{2\sigma^2} \|x - y_t\|^2 \right) \right) dx \\
&\propto \int_{\mathbb{R}^d} e^{-f(x)} \exp\left( -\frac{1}{2\sigma^2} \sum_{t=1}^m \|x - y_t\|^2 \right) dx.
\end{aligned}
$$

The $(\sigma, m)$-density $p(y_{1:m})$ is permutation invariant under the permutation of measurement indices:

$$
\pi : [m] \rightarrow [m].
$$

In the calculation below we derive a general form for $p(y_{1:m})$ where this permutation invariance becomes apparent in terms of the empirical mean of the $m$ measurements

$$
\overline{y}_{1:m} = \frac{1}{m} \sum_{t=1}^m y_t.
$$

and the empirical mean of $\{\|y_t\|^2\}_{t=1}^m$. We start with a rewriting of $\log p(y_{1:m}|x)$:

$$
\begin{aligned}
-2\sigma^2 \log p(y_{1:m}|x) &= \sum_{t=1}^m \|y_t - x\|^2 + \text{cst} \\
&= m\|x\|^2 - 2\langle \sum_{t=1}^m y_t, x \rangle + \sum_{t=1}^m \|y_t\|^2 + \text{cst} \\
&= m\left( \|x - \overline{y}_{1:m}\|^2 - \|\overline{y}_{1:m}\|^2 + \frac{1}{m} \sum_{t=1}^m \|y_t\|^2 \right) + \text{cst},
\end{aligned}
$$

where cst is a constant that does not depend on $y_{1:m}$. Using the above expression, we arrive at:

$$
\begin{aligned}
\log p(y_{1:m}) &= \log \int e^{-f(x)} p(y_{1:m}|x) \, dx + \text{cst} \\
&= \log \int e^{-f(x)} \exp\left( -\frac{m}{2\sigma^2} \|x - \overline{y}_{1:m}\|^2 \right) dx + \frac{\|\overline{y}_{1:m}\|^2 - m^{-1} \sum_{t=1}^m \|y_t\|^2}{2m^{-1}\sigma^2} + \text{cst}.
\end{aligned}
$$

The equation above reduces to (2.2) with the following definition

$$
\varphi(y; \sigma) = \log \int e^{-f(x)} \exp\left( -\frac{1}{2\sigma^2} \|x - y\|^2 \right) dx.
$$

## A.2 DERIVATION FOR (2.4)

Next, we derive the expression for $\hat{x}(y_{1:m}) = \mathbb{E}[X|y_{1:m}]$ given in (2.4):

$$
\begin{aligned}
\mathbb{E}[X|y_{1:m}] &= y_t + \sigma^2 \nabla_{y_t} \log p(y_{1:m}) \\
&= y_t + \sigma^2 \left( m^{-1} \nabla\varphi(\overline{y}_{1:m}; m^{-1/2}\sigma) + \sigma^{-2}(\overline{y}_{1:m} - y_t) \right) \\
&= \overline{y}_{1:m} + m^{-1}\sigma^2 \nabla\varphi(\overline{y}_{1:m}; m^{-1/2}\sigma).
\end{aligned} \tag{A.1}
$$

The first equation above comes from the generalization of the Bayes estimator to factorial kernels (Saremi & Srivastava, 2022), and for the second equation we used (2.2).

## B SMOOTHED SCORE FUNCTIONS

In this section, we give several different expressions related to smoothed densities used in the paper. We consider $Y \sim \mathcal{N}(X, \sigma^2 I)$ where $X \sim e^{-f(x)}$ in $\mathbb{R}^d$, thus with the density

$$p(y) = \int \frac{1}{Z} e^{-f(x)} \frac{1}{(2\pi\sigma^2)^{d/2}} e^{-\frac{1}{2\sigma^2}\|x-y\|^2} dx \propto \int e^{-f(x)} e^{-\frac{1}{2\sigma^2}\|x-y\|^2} dx.$$

We have the following expressions for $\log p(y)$:

$$\log p(y) = \log \int e^{-f(x)} e^{-\frac{1}{2\sigma^2}\|x-y\|^2} dx + \mathrm{cst},$$

$$\log p(y) = \log \int e^{-f(x+y)} e^{-\frac{1}{2\sigma^2}\|x\|^2} dx + \mathrm{cst},$$

leading to

$$\nabla(\log p)(y) = \frac{\int e^{-f(x)} e^{-\frac{1}{2\sigma^2}\|x-y\|^2} \frac{x-y}{\sigma^2} dx}{\int e^{-f(x)} e^{-\frac{1}{2\sigma^2}\|x-y\|^2} dx} = \frac{1}{\sigma^2}(\mathbb{E}[X|y] - y), \tag{B.1}$$

$$\nabla(\log p)(y) = -\frac{\int e^{-f(x+y)} \nabla f(x+y) e^{-\frac{1}{2\sigma^2}\|x\|^2} dx}{\int e^{-f(x+y)} e^{-\frac{1}{2\sigma^2}\|x\|^2} dx}$$

$$= -\frac{\int e^{-f(x)} \nabla f(x) e^{-\frac{1}{2\sigma^2}\|x-y\|^2} dx}{\int e^{-f(x)} e^{-\frac{1}{2\sigma^2}\|x-y\|^2} dx} = -\mathbb{E}[\nabla f(X)|y]. \tag{B.2}$$

This in turn leads to three expressions for the Hessian:

$$\nabla^2(\log p)(y) = -\frac{1}{\sigma^2} I + \frac{1}{\sigma^4}\left(\mathbb{E}\left[x(x-y)^\top|y\right] - \mathbb{E}\left[x|y\right]\mathbb{E}\left[x-y|y\right]^\top\right)$$

$$= -\frac{1}{\sigma^2} I + \frac{1}{\sigma^4}\left(\mathbb{E}\left[xx^\top|y\right] - \mathbb{E}\left[x|y\right]\mathbb{E}\left[x|y\right]^\top\right) = -\frac{1}{\sigma^2} I + \frac{1}{\sigma^4}\mathrm{cov}(X|y), \tag{B.3}$$

$$\nabla^2(\log p)(y) = -\mathbb{E}[\nabla^2 f(X)|y] + \mathbb{E}[\nabla f(X)\nabla f(X)^\top|y] + \mathbb{E}[\nabla f(X)|y]\mathbb{E}[\nabla f(X)|y]^\top$$

$$= -\mathbb{E}[\nabla^2 f(X)|y] + \mathrm{cov}(\nabla f(X)|y), \tag{B.4}$$

$$\nabla^2(\log p)(y) = -\frac{1}{\sigma^2}\left(\mathbb{E}\left[(x-y)\nabla f(x)^\top|y\right] - \mathbb{E}\left[(x-y)|y\right]\mathbb{E}\left[\nabla f(x)|y\right]^\top\right)$$

$$= -\frac{1}{\sigma^2}\left(\mathbb{E}\left[x\nabla f(x)^\top|y\right] - \mathbb{E}\left[x|y\right]\mathbb{E}\left[\nabla f(x)|y\right]^\top\right) = -\frac{1}{\sigma^2}\mathrm{cov}(X, \nabla f(X)|y).$$

### B.1 GAUSSIAN EXAMPLE

If $X \sim \mathcal{N}(\mu, C)$, then $Y \sim \mathcal{N}(\mu, C + \sigma^2 I)$, and then

$$\mathbb{E}[X|Y] = \mu + C(C + \sigma^2 I)^{-1}(Y - \mu),$$

$$\mathrm{cov}(X|Y) = C - C(C + \sigma^2 I)^{-1}C.$$

Therefore, $\mathbb{E}[X|Y]$ is Gaussian with mean $\mu$ and the covariance matrix $C(C + \sigma^2 I)^{-1}C$. We have $-\nabla^2(\log p)(y) = (C + \sigma^2 I)^{-1}$.

### B.2 DISTRIBUTION OF $\mathbb{E}[X|Y]$

If $Y$ is sampled from $p_Y$, let $\hat{p}_\sigma$ be the distribution of $\mathbb{E}[X|Y] = Y + \sigma^2 \nabla(\log p)(Y)$. We can bound the 2-Wasserstein distance between $p$ and $\hat{p}_\sigma$, by sampling $X$ from $p_X$, and taking $\hat{x} = Y + \sigma^2 \nabla \log p(Y)$, where $Y \sim \mathcal{N}(X, \sigma^2 I)$. This leads to a particular "coupling" (Villani, 2021) between $p$ and $\hat{p}_\sigma$. It follows:

$$W_2(p, \hat{p}_\sigma)^2 \leqslant \mathbb{E}_{X,Y}\|\mathbb{E}[X|Y] - X\|^2.$$

Given that the conditional expectation $\mathbb{E}[X|Y]$ is the optimal estimator for the square loss, the last expression is less than the trival estimator $Y$, that is,

$$W_2(p, \hat{p}_\sigma)^2 \leqslant \mathbb{E}_{X,Y}\|Y - X\|^2 = \sigma^2 d. \tag{B.5}$$

### B.3 PROPOSITION 2

The calculations in Sec. B.2 that resulted in (B.5) leads to the following:

*Proof for Proposition 2.* Using Proposition 1, we have

$$W_2(p, \hat{p}_{\sigma,m}) = W_2(p, \hat{p}_{m^{-1/2}\sigma}),$$

which is then combined with (B.5):

$$W_2(p, \hat{p}_{\sigma,m})^2 \leqslant \frac{\sigma^2}{m} d.$$

$\square$

## C PROOFS FOR SEC. 3

### C.1 PROPOSITION 3

*Proof for Proposition 3.* We start with an expression for $\log p(y_{1:m})$:

$$\log p(y_{1:m}) = \log \int_{\mathbb{R}^d} \left( \prod_{k=1}^m \mathcal{N}(x; y_k, \sigma^2 I) \right) \mathcal{N}(x; 0, C) \, dx$$

$$= \sum_{i=1}^d \log \int_{\mathbb{R}} \exp\left( -\sum_{k=1}^m \frac{(y_{ki} - x_i)^2}{2\sigma^2} - \frac{x_i^2}{2\tau_i^2} \right) dx_i + \text{cst} \quad \text{(C.1)}$$

$$= \sum_{i=1}^d \log \int_{\mathbb{R}} \exp\left( -\frac{(x_i - \alpha_i)^2}{2\beta_i^2} - \gamma_i \right) dx_i + \text{cst}.$$

The expressions for $\alpha_i$, $\beta_i$, and $\gamma_i$ are given next by completing the square via matching second, first and zeroth derivative (in that order) of the left and right hand sides below

$$-\sum_{t=1}^m \frac{(y_{ti} - x_i)^2}{2\sigma^2} - \frac{x_i^2}{2\tau_i^2} = -\frac{(x_i - \alpha_i)^2}{2\beta_i^2} - \gamma_i$$

evaluated at $x_i = 0$. The following three equations follow:

$$\frac{1}{\beta_i^2} = \frac{m}{\sigma^2} + \frac{1}{\tau_i^2},$$

$$\frac{\alpha_i}{\beta_i^2} = \sum_{t=1}^m \frac{y_{ti}}{\sigma^2} \Rightarrow \alpha_i = \frac{1}{m + \sigma^2\tau_i^{-2}} \sum_{t=1}^m y_{ti},$$

$$-\frac{\alpha_i^2}{2\beta_i^2} - \gamma_i = -\sum_{t=1}^m \frac{y_{ti}^2}{2\sigma^2} \Rightarrow \gamma_i = \sum_{t=1}^m \frac{y_{ti}^2}{2\sigma^2} - \frac{1}{2\sigma^2(m + \sigma^2\tau_i^{-2})} \left( \sum_{t=1}^m y_{ti} \right)^2.$$

It is convenient to define:

$$A_{ti} = \frac{1}{t + \sigma^2\tau_i^{-2}}. \quad \text{(C.2)}$$

Using above expressions, (C.1) simplifies to:

$$\log p(y_{1:m}) = -\sum_{i=1}^d \gamma_i + \text{cst} = -\sum_{t=1}^m \frac{\|y_t\|^2}{2\sigma^2} + \frac{1}{2\sigma^2} \sum_{i=1}^d A_{mi} \left( \sum_{t=1}^m y_{ti} \right)^2 + \text{cst}. \quad \text{(C.3)}$$

The energy function can be written more compactly by introducing the matrix $F_{\sigma,m}$:

$$\log p(y_{1:m}) = -\frac{1}{2} y_{1:m}^\top F_{\sigma,m} y_{1:m},$$

$$\sigma^2 [F_{\sigma,m}]_{ti,t'i'} = ((1 - A_{mi})\delta_{tt'} - A_{mi}(1 - \delta_{tt'})) \, \delta_{ii'}.$$

In words, the $md \times md$ dimensional matrix $F_{\sigma,m}$ is block diagonal with $d$ blocks of size $m \times m$. The blocks themselves capture the interactions between different measurements indexed by $t$ and $t'$: The $m \times m$ blocks of the matrix $F_{\sigma,m}$, indexed by $i \in [d]$ thus have the form:

$$\sigma^2 F_{\sigma,m}^{(i)} = (1 - A_{mi})I_m + A_{mi}(I_m - 1_m 1_m^\top),$$

where $1_m^\top = (1, 1, \ldots, 1)$ is $m$-dimensional. It is straightforward to find the $m$ eigenvalues of the $m \times m$ matrix $F_{\sigma,m}^{(i)}$ for $i \in [d]$:

- $m - 1$ degenerate eigenvalues equal to $\sigma^{-2}$ corresponding to the eigenvectors

$$\{(1, -1, 0, \ldots, 0)^\top, (1, 0, -1, \ldots, 0)^\top, \ldots, (1, 0, 0, \ldots, -1)^\top\},$$

- one eigenvalue equal to $\sigma^{-2}(1 - mA_{mi})$ corresponding to the eigenvector $(1, 1, \ldots, 1)^\top$.

Since $mA_{mi} > 0$ we have:

$$\lambda_{\max}(F_{\sigma,m}) = \sigma^{-2},$$

which is $(m-1)d$ degenerate. The remaining $d$ eigenvalues are $\{\sigma^{-2}(1 - mA_{mi})\}_{i \in [d]}$, the smallest of which is given by

$$\lambda_{\min}(F_{\sigma,m}) = \sigma^{-2}\left(1 - \frac{m}{m + \sigma^2 \tau_{\max}^{-2}}\right) = \frac{\sigma^{-2}}{1 + m\sigma^{-2}\tau_{\max}^2}.$$

Thus we have:

$$\kappa_{\sigma,m} = \lambda_{\max}(F_{\sigma,m})/\lambda_{\min}(F_{\sigma,m}) = 1 + m\sigma^{-2}\tau_{\max}^2.$$

$\square$

## C.2 PROPOSITION 4

*Proof for Proposition 4.* Using (C.3) and (C.2) we have:

$$-2\sigma^2 \log p(y_{1:t}) = \sum_{k=1}^{t} \|y_k\|^2 - \sum_{i=1}^{d} A_{ti}\left(\sum_{k=1}^{t} y_{ki}\right)^2.$$

Since $\log p(y_t|y_{1:t-1}) = \log p(y_{1:t}) - \log p(y_{1:t-1})$, we have:

$$-2\sigma^2 \log p(y_t|y_{1:t-1}) = \|y_t\|^2 - \sum_{i=1}^{d} A_{ti}y_{ti}\left(y_{ti} + 2\sum_{k=1}^{t-1} y_{ki}\right) + \text{cst}$$

$$= \sum_{i=1}^{d}(1 - A_{ti}) \cdot \left(y_{ti} - \frac{A_{ti}}{1 - A_{ti}}\sum_{k=1}^{t-1} y_{ki}\right)^2 + \text{cst},$$

Therefore the conditional density $p(y_t|y_{1:t-1})$ is the Gaussian $\mathcal{N}(\mu_{t|t-1}, F_{t|t-1}^{-1})$ with a shifted mean

$$\mu_{t|t-1} = \left(\frac{A_{ti}}{1 - A_{ti}}\sum_{k=1}^{t-1} y_{ki}\right)_{i \in [d]},$$

and with an anisotropic, diagonal covariance/precision matrix whose spectrum is given by:

$$\sigma^2 \lambda_i(F_{t|t-1}) = 1 - (t + \sigma^2 \tau_i^{-2})^{-1}.$$

Thus:

$$\kappa_{t|t-1} = \frac{\lambda_{\max}(F_{t|t-1})}{\lambda_{\min}(F_{t|t-1})} = \frac{1 - (t + \sigma^2 \tau_{\min}^{-2})^{-1}}{1 - (t + \sigma^2 \tau_{\max}^{-2})^{-1}}.$$

Lastly, to prove monotonicity result (3.3) we do an analytic continuation of $\kappa_{t|t-1}$ to continuous values by defining $\eta(t) = \kappa_{t|t-1}$ and taking its derivative, below $R = \sigma^2 \tau_{\min}^{-2}, r = \sigma^2 \tau_{\max}^{-2}$, thus

$R > r$:

$$
\begin{aligned}
\eta'(t) &= \frac{(t+R)^{-2}}{1 - (t+R)^{-1}} - \frac{(1 - (t+R)^{-1})(t+r)^{-2}}{(1 - (t+r)^{-1})^2} \\
&= \frac{(t+r)(t+r-1)}{(t+R)^2(t+r-1)^2} - \frac{(t+R-1)(t+R)}{(t+r-1)^2(t+R)^2} \\
&= \frac{(t+r)^2 - r - (t+R)^2 + R}{(t+r-1)^2(t+R)^2} \\
&= \frac{(r-R)(2t+r+R-1)}{(t+r-1)^2(t+R)^2} < 0.
\end{aligned}
$$

$\square$

# D  PROOFS FOR SEC. 4

## D.1  LEMMA 1

*Proof for Lemma 1.* We would like to find an upper bound for $\nabla^2(\log p)(y)$. By using

$$
\nabla^2(\log p)(y) = -\sigma^{-2}I + \sigma^{-4}\mathrm{cov}(X|y),
$$

derived in Appendix B, we need to upper bound $\mathrm{cov}(X|y)$. It suffices to study $\mathbb{E}[\|X - x_0\|^2|y]$ since

$$
\mathrm{cov}(X|y) = \mathrm{cov}(X - x_0|y) \preccurlyeq \mathbb{E}[\|X - x_0\|^2|y]\,I.
$$

We have by a convexity argument[5]:

$$
\begin{aligned}
\mathbb{E}[\|X - x_0\|^2|y] &= \frac{\int e^{-f(x) - \frac{1}{2\sigma^2}\|x-y\|^2}\|x - x_0\|^2 dx}{\int e^{-f(x) - \frac{1}{2\sigma^2}\|x-y\|^2} dx} \\
&\leqslant \frac{\int e^{-f(x) + \frac{1}{\sigma^2}(x-x_0)^\top(y-x_0)}\|x - x_0\|^2 dx}{\int e^{-f(x) + \frac{1}{\sigma^2}(x-x_0)^\top(y-x_0)} dx}.
\end{aligned}
\tag{D.1}
$$

Next, we find an upper bound for $\|x - x_0\|^2$ itself. Using the assumption in the lemma we have:

$$
\|\nabla f(x) + \sigma^{-2}(x_0 - y)\| \geqslant \|\nabla f(x)\| - \sigma^{-2}\|x_0 - y\| \geqslant \mu\|x - x_0\| - \Delta - \sigma^{-2}\|x_0 - y\|,
$$

leading to $\mu\|x - x_0\| \leqslant \|\nabla f(x) + \sigma^{-2}(x_0 - y)\| + \Delta + \sigma^{-2}\|x_0 - y\|$ and thus

$$
\mu^2\|x - x_0\|^2 \leqslant 3\|\nabla f(x) + \frac{x_0 - y}{\sigma^2}\|^2 + 3\Delta^2 + 3\frac{\|x_0 - y\|^2}{\sigma^4}.
$$

Finally, using (D.1), we only need to find an upper-bound for $\|\nabla f(x) + \sigma^{-2}(x_0 - y)\|^2$ under the distribution $\tilde{p}(x) \propto e^{-\tilde{f}(x)}$, where $\tilde{f}(x) = f(x) - \sigma^{-2}(x - x_0)^\top(y - x_0)$. This is achieved with:

$$
\int \tilde{p}(x)\|\nabla f(x) + \frac{x_0 - y}{\sigma^2}\|^2 dx = \int \tilde{p}(x)\|\nabla\tilde{f}(x)\|^2 dx = \int \tilde{p}(x)\,\mathrm{tr}\,\nabla^2 f(x)dx \leqslant Ld,
$$

where second equality is obtained using integration by parts akin to score matching (Hyvärinen, 2005), and for the last inequality we used our assumption $\nabla^2 f(x) \preccurlyeq LI$. Putting all together we arrive at:

$$
\nabla^2(\log p)(y) \preccurlyeq \left(-1 + \frac{3Ld}{\mu^2\sigma^2} + \frac{3\Delta^2}{\mu^2\sigma^2} + \frac{3\|x_0 - y\|^2}{\mu^2\sigma^6}\right)\frac{I}{\sigma^2}.
$$

$\square$

---

[5]We consider the exponential family

$$
p(x|\nu) = \exp\left(-f(x) + \frac{1}{\sigma^2}(x - x_0)^\top(y - x_0) - \frac{\nu}{2\sigma^2}\|x - x_0\|^2 - \frac{\nu}{2\sigma^2}\|y - x_0\|^2 - a(\nu)\right).
$$

Since $a(\nu)$ is convex (Wainwright & Jordan, 2008), we have $a'(1) \geqslant a'(0)$, which is exactly the desired statement.

## D.2 LEMMA 2

Since $\log p(y_t|y_{1:t-1}) = \log p(y_{1:t}) - \log p(y_{1:t-1})$, we have

$$\nabla_{y_t}^k \log p(y_t|y_{1:t-1}) = \nabla_{y_t}^k \log p(y_{1:t}).$$

Start with the score function ($k = 1$):

$$\nabla_{y_t} \log p(y_{1:t}) = \frac{\int \sigma^{-2}(x - y_t)p(x) \prod_{i=1}^t p(y_i|x)dx}{\int p(x) \prod_{i=1}^t p(y_i|x)dx} = \sigma^{-2}\left(\mathbb{E}[X|y_{1:t}] - y_t\right).$$

Next, we derive the Hessian $\nabla_{y_t}^2 \log p(y_{1:t}) = -\sigma^{-2}I + A_t - B_t$, where $A_t$ and $B_t$ are given by:

$$A_t = \frac{\int \sigma^{-4}(x - y_t)(x - y_t)^\top p(x) \prod_{i=1}^t p(y_i|x)dx}{\int p(x) \prod_{i=1}^t p(y_i|x)dx} = \sigma^{-4}\,\mathbb{E}[(X - y_t)(X - y_t)^\top|y_{1:t}],$$

$$B_t = \sigma^{-4}\left(\frac{\int (x - y_t)p(x) \prod_{i=1}^t p(y_i|x)dx}{\int p(x) \prod_{i=1}^t p(y_i|x)dx}\right)\left(\frac{\int (x - y_t)p(x) \prod_{i=1}^t p(y_i|x)dx}{\int p(x) \prod_{i=1}^t p(y_i|x)dx}\right)^\top$$

$$= \sigma^{-4}\left(\mathbb{E}[X|y_{1:t}] - y_t\right)\left(\mathbb{E}[X|y_{1:t}] - y_t\right)^\top.$$

By simplifying $A_t - B_t$, the $y_t$ cross-terms cancel out and the posterior covariance matrix emerges:

$$\nabla_{y_t}^2 \log p(y_{1:t}) = -\sigma^{-2}I + \sigma^{-4}\mathrm{cov}(X|y_{1:t}). \tag{D.2}$$

The lemma is proven since the mean of the posterior covariance $\mathbb{E}_{y_{1:t}}\mathrm{cov}(X|y_{1:t})$ can only go down upon accumulation of measurements (conditioning on more variables).

## D.3 THEOREM 1

*Proof for Theorem 1.* We start with the definition of $(\sigma, m)$-density:

$$p(y_{1:m}) = \int_{\mathcal{Z}} p(z)p(y_{1:m}|z)dz \propto \int_{\mathcal{Z}} p(z)\left(\int_{\mathcal{X}} \exp\left(-\frac{1}{2\tau^2}\|x - z\|^2 - \frac{1}{2\sigma^2}\sum_{t=1}^m \|x - y_t\|^2\right)dx\right)dz,$$

which we express by integrating out $x$. We have

$$p(y_{1:m}|z) \propto \int_{\mathcal{X}} \exp\left(-\frac{1}{2\tau^2}\|x - z\|^2 - \frac{m}{2\sigma^2}\left(\|x - \overline{y}_{1:m}\|^2 - \|\overline{y}_{1:m}\|^2 + \frac{1}{m}\sum_{t=1}^m \|y_t\|^2\right)\right)dx$$

$$\propto \exp\left(-\frac{m}{2(m\tau^2 + \sigma^2)}\|z - \overline{y}_{1:m}\|^2 + \frac{m}{2\sigma^2}\|\overline{y}_{1:m}\|^2 - \frac{1}{2\sigma^2}\sum_{t=1}^m \|y_t\|^2\right).$$

We can then express $\nabla_{y_m} \log p(y_{1:m})$ in terms of $\mathbb{E}[Z|y_{1:m}]$:

$$\nabla_{y_m} \log p(y_{1:m}) = \frac{m\tau^2}{\sigma^2(m\tau^2 + \sigma^2)}\overline{y}_{1:m} - \frac{y_m}{\sigma^2} + \frac{1}{m\tau^2 + \sigma^2}\mathbb{E}[Z|y_{1:m}].$$

Next, we take another derivative:

$$\nabla_{y_m}^2 \log p(y_{1:m}) = \left(\frac{\tau^2}{\sigma^2(m\tau^2 + \sigma^2)} - \frac{1}{\sigma^2}\right) \cdot I + \frac{1}{m\tau^2 + \sigma^2}\nabla_{y_m}\mathbb{E}[Z|y_{1:m}].$$

Next, we compute $\nabla_{y_m}\mathbb{E}[Z|y_{1:m}]$:

$$\nabla_{y_m}\mathbb{E}[Z|y_{1:m}] = \frac{1}{m\tau^2 + \sigma^2}\mathbb{E}[ZZ^\top|y_{1:m}] + \frac{m\tau^2}{m\tau^2 + \sigma^2}\mathbb{E}[Z|y_{1:m}]\overline{y}_{1:m}^\top - \frac{1}{\sigma^2}\mathbb{E}[Z|y_{1:m}]y_m^\top$$

$$- \mathbb{E}[Z|y_{1:m}]\left(\frac{1}{m\tau^2 + \sigma^2}\mathbb{E}[Z|y_{1:m}] + \frac{m\tau^2}{m\tau^2 + \sigma^2}\overline{y}_{1:m} - \frac{1}{\sigma^2}y_m\right)^\top$$

$$= \frac{1}{m\tau^2 + \sigma^2}\mathrm{cov}(Z|y_{1:m}).$$

Putting all together, we arrive at:[6]

$$\nabla^2_{y_m} \log p(y_{1:m}) = \frac{1}{\sigma^2}\Big(\frac{\tau^2}{m\tau^2 + \sigma^2} - 1\Big)\cdot I + \frac{1}{(m\tau^2 + \sigma^2)^2}\mathrm{cov}(Z|y_{1:m}). \tag{D.4}$$

Finally, since $\|Z\|^2 \leqslant R^2$ almost surely, we have $\mathrm{cov}(Z|y_{1:m}) \preccurlyeq R^2 I$, therefore

$$\nabla^2_{y_m} \log p(y_m|y_{1:m-1}) = \nabla^2_{y_m} \log p(y_{1:m}) \preccurlyeq \zeta(m)I,$$

where

$$\zeta(m) = \frac{1}{\sigma^2}\Big(\frac{\tau^2}{m\tau^2 + \sigma^2} - 1\Big) + \frac{R^2}{(m\tau^2 + \sigma^2)^2}.$$

$\square$

### D.4 DERIVATION FOR (4.6)

In this example, we have $X = Z + N_0$, $N_0 \sim \mathcal{N}(0, \tau^2 I)$, and $Y = X + N_1$, $N_1 \sim \mathcal{N}(0, \sigma^2 I)$, therefore

$$Y = Z + N, \ N \sim \mathcal{N}(0, (\sigma^2 + \tau^2)I), \tag{D.5}$$

where $Z \sim p(z)$,

$$p(z) = \frac{1}{2}\delta(z - \mu) + \frac{1}{2}\delta(z + \mu),$$

$\delta$ is the Dirac delta function in $d$-dimensions. Alternatively, we have $p(y) = \int p(z)p(y|z)dz$, where

$$p(y|z) = \mathcal{N}(y; z, (\sigma^2 + \tau^2)I).$$

Using (B.3), adapted for (D.5), we have:

$$H(y) = -\nabla^2(\log p)(y) = \frac{1}{\sigma^2 + \tau^2}\Big(I - \frac{1}{\sigma^2 + \tau^2}\mathrm{cov}(Z|y)\Big). \tag{D.6}$$

Next, we derive an expression for $\mathrm{cov}(Z|y)$:

$$\mathrm{cov}(Z|y) = \mathbb{E}[ZZ^\top|y] - \mathbb{E}[Z|y]\mathbb{E}[Z|y]^\top.$$

We have:

$$\mathbb{E}[ZZ^\top|y] = \mu\mu^\top,$$
$$\mathbb{E}[Z|y] = \mu \cdot \frac{e^{-A} - e^{-B}}{e^{-A} + e^{-B}},$$
$$A = \frac{\|y - \mu\|^2}{2(\sigma^2 + \tau^2)},$$
$$B = \frac{\|y + \mu\|^2}{2(\sigma^2 + \tau^2)}.$$

It follows:

$$\mathrm{cov}(Z|y) = \mu\mu^\top \cdot \Big(1 - \Big(\frac{e^{-A} - e^{-B}}{e^{-A} + e^{-B}}\Big)^2\Big) = \frac{2\mu\mu^\top}{1 + \cosh(B - A)}$$
$$= 2\mu\mu^\top \cdot \Big(1 + \cosh\Big(\frac{2\mu^\top y}{\sigma^2 + \tau^2}\Big)\Big)^{-1}. \tag{D.7}$$

By combining (D.6) and (D.7) we arrive at (4.6).

---

[6]Not required for the proof, but we can also relate $\mathrm{cov}(X|y_{1:m})$ and $\mathrm{cov}(Z|y_{1:m})$ directly since as we know (see the proof of Lemma 2):

$$\nabla^2_{y_m} \log p(y_{1:m}) = -\frac{1}{\sigma^2}I + \frac{1}{\sigma^4}\mathrm{cov}(X|y_{1:m}). \tag{D.3}$$

By combining (D.4) and (D.3), it follows:

$$\mathrm{cov}(X|y_{1:m}) = \frac{\sigma^2\tau^2}{m\tau^2 + \sigma^2}I + \Big(\frac{\sigma^2}{m\tau^2 + \sigma^2}\Big)^2 \mathrm{cov}(Z|y_{1:m}).$$

# E    DETAILED ALGORITHM

---

**Algorithm 2:** MCMC$_\sigma$ in Algorithm 1 via Underdamped Langevin MCMC by Sachs et al. (2017)

---

1: **Input** $Y_t^{(i-1)}, \overline{Y}_{1:t-1}$
2: **Parameters** current MCMC iteration $i$, current measurement index $t$, step size $\delta$, friction $\gamma$, steps taken $n_t$, estimated smoothed score function $\hat{g}(y; \sigma)$, Lipschitz parameter $L$, noise level $\sigma$
3: **Output** $Y_t^{(i)}$
4: Initialize $Y_t^{(i,0)} \sim \text{Unif}([0,1]^d) + \mathcal{N}(0, \sigma^2 I)$
5: Initialize $V \leftarrow 0$
6: **for** $k = [0, \ldots, K-1]$ **do**
7:     $Y_t^{(i,k+1)} = Y_t^{(i,k)} + \frac{\delta}{2} V$
8:     $\overline{Y}_{1:t} = \overline{Y}_{1:t-1} + (Y_t^{(i,k+1)} - \overline{Y}_{1:t-1})/t$
9:     $G = t^{-1} \hat{g}(\overline{Y}_{1:t}; t^{-1/2}\sigma) + \sigma^{-2}(\overline{Y}_{1:t} - Y_t^{(i,k+1)})$ according to (4.9)
10:     $V \leftarrow V + \frac{\delta}{2L} G$
11:     $B \sim \mathcal{N}(0, I)$
12:     $V \leftarrow \exp(-\gamma\delta) V + \frac{\delta}{2L} G + \sqrt{\frac{1}{L}(1 - \exp(-2\gamma\delta))} B$
13:     $Y_t^{(i,k+1)} \leftarrow Y_t^{(i,k+1)} + \frac{\delta}{2} V$
14: **end for**
15: **return** $Y_t^{(i)} = Y_t^{(i,K)}$

---

# F    ESTIMATORS FOR $\nabla \log p(y)$

The purpose of this section is to derive estimators for the smoothed score function

$$g(y; \sigma) = \nabla(\log p)(y) = \nabla\varphi(y; \sigma),$$

which can be used to run Algorithm 2, in turn running Algorithm 1. We first derive (4.8) by rewriting (B.1) as follows:

$$g(y; \sigma) = \frac{1}{\sigma^2} \frac{\int (x-y) \exp(-\check{f}(x; y, \sigma)) dx}{\int \exp(-\check{f}(x; y, \sigma)) dx} = \frac{1}{\sigma^2}(\mathbb{E}[\check{X}] - y), \quad \check{X} \sim e^{-\check{f}(\cdot; y, \sigma)},$$

where

$$\check{f}(x; y, \sigma) := f(x) + \frac{\|x-y\|^2}{2\sigma^2}.$$

## F.1    THE PLUG-IN ESTIMATOR $\hat{g}_{\text{plugin}}$

We can arrive at a simple plug-in estimator for $g(y; \sigma)$ by rewriting (B.1) as follows:

$$\nabla\varphi(y; \sigma) = \frac{1}{\sigma^2} \frac{\mathbb{E}_{N \sim \mathcal{N}(0; \sigma^2 I)}[N e^{-f(N+y)}]}{\mathbb{E}_{N \sim \mathcal{N}(0; \sigma^2 I)}[e^{-f(N+y)}]},$$

We then simply estimate the numerator and denominator above using i.i.d. Gaussian draws:

$$\hat{g}_{\text{plugin}}(y; \sigma) = \frac{1}{\sigma} \frac{\sum_{i=1}^{n} \varepsilon_i \exp(-f(\sigma\varepsilon_i + y))}{\sum_{i=1}^{n} \exp(-f(\sigma\varepsilon_i + y))}, \quad \varepsilon_i \overset{\text{iid}}{\sim} \mathcal{N}(0, I). \tag{F.1}$$

(Above, we used one set of i.i.d. draws for estimating the numerator and denominator used in our experiments. Of course, one should take independent draws if computation budget is not an issue.)

**Gradient-based plug-in estimator.**    One can also obtain a different type of plug-in estimator for $\nabla \log p(y)$ using (B.2) that directly takes the gradient information $\nabla f(x)$ into consideration. This estimator should have better properties, but we did not experiment with it in this paper. Intuitively, such plug-in estimators (gradient-aware or not) will suffer from the curse of dimensionality.

**Connections to importance sampling.** One can also arrive at (F.1) by using importance sampling to estimate $\mathbb{E}[\check{X}]$ in (4.8), which is insightful. In particular, in importance sampling instead of sampling from the probability measure $\nu$ associated with $\exp(-\check{f}(\cdot; y, \sigma))$ (which is hard to sample from) we sample from an easier probability measure $\mu$, which in our case we took it to be the Gaussian measure $\mathcal{N}(y; \sigma^2 I)$. With this setup, the estimator (F.1) follows through. It is known that the number of samples required to have an accurate estimator based on importance sampling is of order $\exp(\mathrm{KL}(\nu \| \mu))$, where $\mathrm{KL}(\nu \| \mu)$ is the Kullback-Leibler divergence of $\mu$ from $\nu$ (Chatterjee & Diaconis, 2018). This further highlights the limitations of the estimator (F.1).

### F.2 THE LANGEVIN ESTIMATOR $\hat{g}_{\mathrm{langevin}}$

A better way to estimate $\mathbb{E}[\check{X}]$ in (4.8) is to use MCMC — in particular, the gradient-based Langevin MCMC. For any $y$, this is done by running Langevin MCMC using the score function

$$\check{g}(x; y, \sigma) = -\nabla f(x) - \frac{x - y}{\sigma^2}.$$

As an example, and to be concrete, for any $y$ one can sample $\check{X}$ by discretizing the overdamped Langevin diffusion:

$$d\check{X}_s = \left(-\nabla f(\check{X}_s) - \frac{\check{X}_s - y}{\sigma^2}\right) ds + \sqrt{2} dB_s.$$

Given $n$ such independent samples, $(\check{X}_i)_{i=1}^n$ drawn by running the Langevin dynamics, we then use (4.8) to arrive at $\hat{g}_{\mathrm{langevin}}$:

$$\hat{g}_{\mathrm{langevin}}(y; \sigma) = \frac{1}{\sigma^2} \cdot \left(\frac{1}{n} \sum_{i=1}^n \check{X}_i - y\right). \tag{F.2}$$

The procedure above for drawing a sample $\check{X}$ from $e^{-\check{f}(x; y, \sigma)}$ using overdamped Langevin MCMC is given in Algorithm 3.

---

**Algorithm 3:** Draw $\check{X} \sim e^{-\check{f}(x; y, \sigma)}$ with Langevin MCMC for the estimator (F.2)

---

1: **Parameter** noise level $\sigma$
2: **Input** current $y$ for which we need to estimate $g(y; \sigma)$
3: **Hyperparameters** step size $\delta$, number of iterations $K$
4: **Output** $\check{X}$
5: $x_0 = y$
6: **for** $k = [1, \ldots, K]$ **do**
7:    $\check{g} = -\nabla f(x_{k-1}) - \sigma^{-2}(x_{k-1} - y)$
8:    $\varepsilon \sim \mathcal{N}(0, I)$
9:    $x_k = x_{k-1} + \delta \check{g} + \sqrt{2\delta}\, \varepsilon$
10: **end for**
11: **return** $\check{X} \leftarrow x_K$

---

We should highlight that in our experiments, instead of using the overdamped Langevin MCMC above, we used the more sophisticated underdamped Langevin MCMC algorithm by Sachs et al. (2017).

**Connections to "Entropy SGD".** The problem of estimating smoothed score functions have also been of interest in the neural network optimization literature under the terminology of entropy SGD (Chaudhari et al., 2017). There, the smoothed score function (although the kernel smoothing lexicon is not used there; $1/\sigma^2$ is denoted by $\gamma$ and is referred to as *scope*) is utilized for optimization, *not for sampling*, to arrive at flatter minima of the loss landscape with better generalization properties. Due to different motivations, Algorithm 3 differs from the one used in entropy SGD, e.g., we do not have exponential averaging here.

## G EXPERIMENTAL DETAIL

The hyperparameters were tuned on a log-spaced grid. We searched the step size $\delta$ over $\{0.03, 0.1, 0.3, 1.0\}$, the effective friction $\gamma\delta$ over $\{0.0625, 0.125, 0.25, 0.5, 1.0\}$, per-$t$ MCMC iterations $n_t$ over $\{1, 4, 16\}$, and the Lipschitz parameter over $\{1/\sigma^2, 1.0\}$. We found that the hyperparameter combinations ($\delta = 0.03$, $\gamma\delta = 0.0625$) and ($\delta = 1.0$, $\gamma\delta = 0.5$) worked well for most configurations of test density type, $\sigma$, $d$, and MCMC algorithm. For JMS, $m$ ran over $\{200, 400, 600, 800, 1000\}$.

We experimented with two initialization schemes in SMS. At each $t$, the walkers sampling from $p(y_t|y_{1:t-1})$ were initialized at (i) *warm*: $\mathbb{E}[X|y_{1:t-1}] + \varepsilon$, where $\varepsilon \sim \mathcal{N}(0, \sigma^2 I)$, or (ii) *cold*: at samples from $\text{Unif}([-1, 1]^d) + \mathcal{N}(0, \sigma^2 I)$. For convergence experiments, we report results from the warm start, as it was more robust to the choice of hyperparameters. The tunneling results were obtained with cold start. The hyperparameters were tuned for each algorithm while the total number of iterations was kept fixed at a large value. We define each iteration as an MCMC update step. For SMS, the total number of iterations is $\sum_{t=1}^m n_t$, the number of MCMC iterations for each measurement $t$, $n_t$, summed up over the $m$ measurements. We had $n_t \in \{1, 4, 16\}$ and $m = 5,000$. For the remaining three sampling schemes (JMS, $m = 1$, and $\sigma = 0$), the total number of iterations is simply the number of MCMC iterations, fixed to $20,000$, but we found these algorithms to converge much earlier, around $5,000$.

## H MCMC ALGORITHMS

Our algorithm is agnostic to the choice of MCMC sampling algorithm used in the Markovian phases. In this section, we run SMS sampling with four different Langevin MCMC algorithms. The results presented earlier in Sec. 5 uses an ULD algorithm with an Euler discretization scheme that extends the BAOAB integration using multiple time steps for the O-part ("Sachs et al.") (Sachs et al., 2017). Next, we consider two algorithms that operate on the integral representations of ULD. Recall that continuous-time ULD is represented by the following stochastic differential equation (SDE):

$$dv_t = -\gamma v_t dt - u\nabla f(x_t)dt + (\sqrt{2\gamma u})dB_t,$$
$$dx_t = v_t dt,$$

where $x_t, v_t \in \mathbb{R}^d$ and $B_t$ is the standard Brownian motion in $\mathbb{R}^d$. The solution $(x_t, v_t)$ to the continuous-time ULD is

$$v_t = v_0 e^{-\gamma t} - u \left( \int_0^t \exp\left(-(t-s)\right) \nabla f(x_s)ds \right) + \sqrt{2\gamma u} \int_0^t \exp\left(-\gamma(t-s)\right) dB_s,$$

$$x_t = x_0 + \int_0^t v_s ds. \tag{H.1}$$

Similarly, the discrete ULD is defined by the SDE

$$d\tilde{v}_t = -\gamma v_t dt - u\nabla f(\tilde{x}_0)dt + (\sqrt{2\gamma u})dB_t,$$
$$d\tilde{x}_t = \tilde{v}_t dt,$$

which yields the solution

$$\tilde{v}_t = \tilde{v}_0 e^{-\gamma t} - u \left( \int_0^t \exp\left(-(t-s)\right) \nabla f(\tilde{x}_0)ds \right) + \sqrt{2\gamma u} \int_0^t \exp\left(-\gamma(t-s)\right) dB_s,$$

$$\tilde{x}_t = \tilde{x}_0 + \int_0^t \tilde{v}_s ds. \tag{H.2}$$

Shen & Lee (2019) seeks a lower discretization error by using a 2-step fixed point iteration method, or the randomized midpoint method. The integral in (H.1) is evaluated along uniform random points between 0 and $t$. On the other hand, Cheng et al. (2018) computes the moments of the joint Gaussian over $(\tilde{x}_t, \tilde{v}_t)$ in the updates of (H.2). In our comparison we additionally include MALA, an Euler discretization of the overdamped Langevin dynamics represented by the SDE

$$dx_t = -u\nabla f(x_t)dt + (\sqrt{2\gamma u})dB_t, \tag{H.3}$$

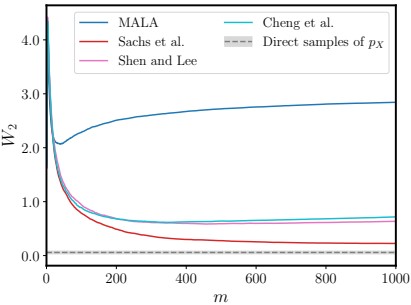

Figure 5: $W_2$ vs. $m$ for various MCMC$_\sigma$ algorithms used in the inner loop of Algorithm 1.

accompanied by Metropolis adjustment to correct for the discretization errors (Roberts & Tweedie, 1996).

The algorithms are compared in Fig. 5 for the Gaussian mixture test density introduced in Sec. 5.2 with $d = 8$. The three (unadjusted) ULD algorithms converge faster than does MALA to a lower $W_2$. In ULD algorithms, Brownian motion affects the positions $x_t$ through the velocities $v_t$, rather than directly as in MALA, resulting in a smoother evolution of $x_t$ that lends itself better to discretization. The first two dimensions of the final samples are displayed in Fig. 6. MALA samples fail to separate into the two modes, whereas Cheng et al, Shen and Lee, and Sachs et al have better sample quality, with Sachs et al performing the best and almost approaching the sample variance of $W_2$ when samples are directly drawn from $p_X$ (the gray band).

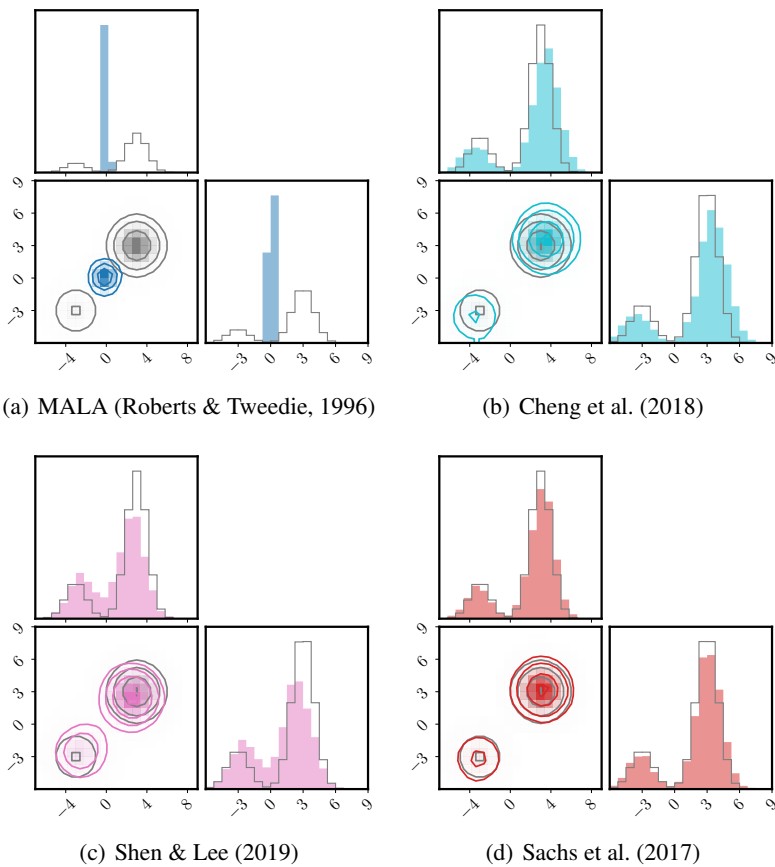

(a) MALA (Roberts & Tweedie, 1996)

(b) Cheng et al. (2018)

(c) Shen & Lee (2019)

(d) Sachs et al. (2017)

Figure 6: Final $\hat{X}$ samples for various MCMC$_\sigma$ algorithms used in the inner loop of Algorithm 1.

## I  SCORE ESTIMATION

Langevin MCMC requires the smoothed score function $g(y; \sigma) = \nabla \log p(y)$. Results presented earlier in Sec. 5 assumed access to the analytic score function. Here, we use the plug-in estimator $\hat{g}_{\text{plugin}}$ presented in (F.1) with varying numbers of MC samples $n$ as well as the Langevin estimator $\hat{g}_{\text{langevin}}$ presented in (F.2).

To prevent numerical underflow, we implemented (F.1) as follows:

$$A = \mathsf{logsumexp}_{i=1}^{n}\big(-f(y + \sigma \varepsilon_i)\big), \tag{I.1}$$

$$B_+ = \mathsf{logsumexp}_{j=1}^{n_+}\big(\log \varepsilon_j - f(y + \sigma \varepsilon_j)\big), \tag{I.2}$$

$$B_- = \mathsf{logsumexp}_{k=1}^{n_-}\big(\log(-\varepsilon_k) - f(y + \sigma \varepsilon_k)\big), \tag{I.3}$$

$$\hat{g}_{\text{plugin}}(y; \sigma) = \frac{1}{\sigma} \cdot \Big(e^{B_+ - A} - e^{B_- - A}\Big), \tag{I.4}$$

where

$$\mathsf{logsumexp}_{i=1}^{n}(a_i) \coloneqq a_{\max} + \log \sum_{i=1}^{n} \exp(a_i - a_{\max}),$$

$a_{\max} \coloneqq \max_i a_i$, and $\varepsilon \sim \mathcal{N}(0, I)$. In (I.2) and (I.3), $j = 1, \ldots, n_+$ and $k = 1, \ldots, n_-$ denote the indices for which $\varepsilon$ is positive and negative, respectively, with $n_+ + n_- = n$. Note that the same Gaussian samples $\varepsilon$ were used to evaluate the numerator and the denominator.

For a Gaussian mixture density introduced in Sec. 5.2 with $d = 2$, both the analytic and the $\hat{g}_{\text{plugin}}$ score functions converge to a low $W_2$, as shown in Fig. 7 (a, b). The sample quality is on par with the analytic score function with $n$ as small as 500 and there is little benefit to increasing the $n$ past 500. The $\hat{g}_{\text{langevin}}$ score function struggles in this low-dimensional example, but catches up to the highest-$n$ $\hat{g}_{\text{plugin}}$ score function at $d = 8$, as Fig. 7 (c, d) shows. All estimated score functions significantly underperform the analytic score. This is a preliminary result: for obtaining $\hat{g}_{\text{langevin}}$ we did not extensively tune the hyperparameters. In addition, for $\hat{g}_{\text{plugin}}$, variance reduction techniques, such as importance weighting, may help get more mileage from finite $n$.

## J  MIXTURE OF CORRELATED GAUSSIANS

In this section, we study a correlated test density, namely a mixture of two 2-dimensional Gaussians with full covariances:

$$p(x) = \alpha \, \mathcal{N}(x; \mu, \Sigma_0) + (1 - \alpha) \, \mathcal{N}(x; -\mu, \Sigma_1). \tag{J.1}$$

We choose $\mu = 3 \cdot 1_d$, $\Sigma_0 = R \, \text{diag}(\frac{1}{4}, 4) \, R^T$, $\Sigma_1 = R^T \, \text{diag}(1, 9) \, R$, $\alpha = \frac{4}{5}$, and the rotation matrix

$$R = \begin{pmatrix} \cos \theta & -\sin \theta \\ \sin \theta & \cos \theta \end{pmatrix}$$

with $\theta = \pi/360$. Fig. 8 compares the performance of the same Langevin MCMC algorithm (Sachs et al., 2017) used within the SMS scheme (a) with that used without Gaussian smoothing (b). In both cases here, we initialized the samplers at the origin (5000 particles total). The total number of iterations are the same between two algorithms (see Appendix G for details).

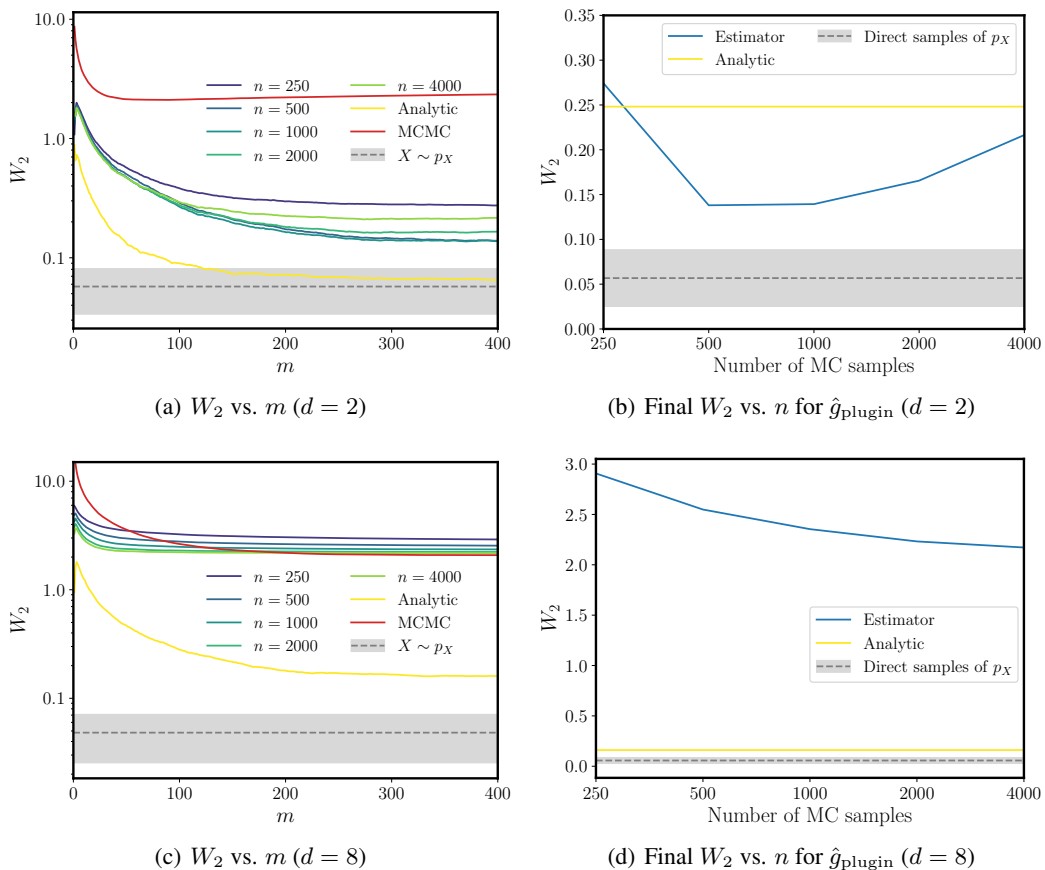

Figure 7: Comparison of the analytic $g(y; \sigma) = \nabla \log p(y)$, the plug-in estimator $\hat{g}_{\text{plugin}}$ (F.1) with varying numbers of MC samples $n$, and the Langevin estimator $\hat{g}_{\text{langevin}}$ (F.2). The test density was the mixture of Gaussians introduced in Sec. 5.2 with $d = 2$ (a, b) and $d = 8$ (c, d).

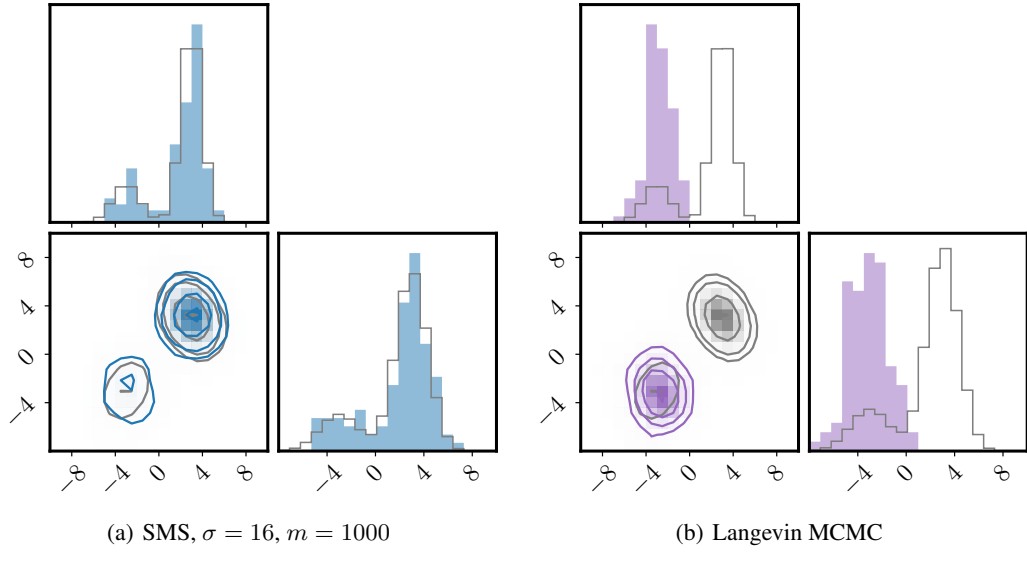

Figure 8: Mixture of correlated Gaussians.

