# OpenReview forum: "Chain of Log-Concave Markov Chains"
_ICLR.cc/2024/Conference — ICLR 2024 poster_

### Official Review · Reviewer_3oQn · 2023-10-30

**Soundness:** 4 excellent
**Presentation:** 4 excellent
**Contribution:** 3 good
**Rating:** 8
**Confidence:** 3

**Summary:**

This paper introduces a general framework for sampling  unnormalized probability densities. The main idea is based on smoothing the distribution
by adding Gaussian noise to the original sample. The process then repeats $m$ times,
each time independently adding gaussian noise to the original sample.
The final step involves computing the Bayes optimal predictor of the original sample based on the i.i.d. samples from the smoothed density.

The authors first demonstrate via an example that sampling all the smoothed observations jointly at once might not be optimal, since the
sampling problem might become ill-conditioned as $m$ increases, which prohibits the use of standard MCMC methods. Subsequently, they demonstrate
that if instead sequential sampling is used, where each sample is taken conditional on the previous samples, then the condition number
of the sampling problem can only improve as $m$ increases, enabling the use of MCMC methods. They prove that this process succeeds to sample
in a general setting where the initial distribution is supported on a bounded subset of $\mathbb{R}^d$. Additionally, they evaluate their
approach experimentally, by sampling from elliptical gaussians (which are ill-conditioned) and mixtures of gaussians.
The performance is then compared to that of standard Langevin MCMC algorithms without smoothing, showing significant benefits both
in terms of distributional convergence, as well as finding less dominant modes of the distribution.

**Strengths:**

This is a very original work, introducing an elegant and intuitive general framework for sampling from unnormalized probability densities. The idea of smoothing
and walk-jump sampling has already appeared in prior work (Saremi & Hyvarinen, 2019), but this is the first time it has been applied to
the general problem of sampling from non-log concave densities. I like the intuition given by the authors about how smoothing helps by "filling in"
probability mass between modes of the original distribution, thus reducing the general problem to the log-concave setting.
One strength of this new approach is that the sequential sampling can simply be implemented by keeping a running average of all the previous
samples up until this point, so the memory footprint is small.
Moreover, this paper is written with great clarity, as all theoretical results are explained in detail and they convincingly demonstrate the power of the new approach.
The experiments also complement the theory very nicely, by showing the benefit of running Langevin with smoothing. The tunneling phenomena that
are observed from the simulations are also very interesting. Lastly, most of the proofs follow from standard techniques, but they are at an
adequate level for ICLR.

**Weaknesses:**

One thing that is missing from this paper is the final step of computing the bayes estimator $\mathbb{E}[X|y_{1:m}]$ to obtain samples from the true
distribution.
For that, it is needed to estimate the score function $\nabla \log p(y_{1:m})$ of the smoothed distribution. The authors acknowledge that this is
a separate issue that needs to be addressed. I understand this might be beyond the scope of the current work, which is more focused on
showing the benefits of smoothing. However, ultimately estimating the score function is a crucial step for this to be an end-to-end approach
for sampling and it is not clear what is the difficulty of this problem compared to the original sampling problem (is it strictly easier in some sense?).
In my opinion, this would make clear what the benefit of this approach over standard MCMC algorithms is.

**Questions:**

-Related to a previous point about estimating the score function, how exactly do the authors arrive at the expression (4.8) for estimating the
score function? In appendix B, a number of properties is proven about the score function, among which that it is related to $\mathbb{E}[X|y_{1:m}]$.
However, I don't see how this is captured in the empirical definition of (4.8). What is the meaning of taking a weighted average of
the $\epsilon_i$s?

-As the authors discuss in Remark 2, the convergence of the condition number to $1$ with rate $1/m$ holds as long as the original distribution involves
some (even small) additive gaussian noise. Why is it enough that even a tiny bit of Gaussian noise be added to the distribution for this decay to happen?
Do the authors have some intuition about that statement?

-Related to the previous question, if we care about the expected hessian of the conditional sampling, I'm wondering whether it's possible to have a version of Theorem 1 showing that the expected condition number of the conditional sampling problem
converges to $1$ with rate $1/m$ without assuming that the original distribution contains some amount of Gaussian noise. The reason is
the following: the proof for all the claims about the "increase" in log-concavity basically hinge on equation (D.2), which asserts that
$\nabla^2 \log p(y_{1:m}) = -\sigma^{-2} I + \sigma^{-4} Cov(X|y_{1:m})$. The argument then proceeds by the observation that conditioning on more and
more observations can only reduce the variance of $X$ in expectation, so the "positive" part on the right hand side can only become smaller as
$m$ increases. The proof of Theorem 1 expresses $Cov(X|y_{1:m})$ as $O(1/ m) + O(1/m^2) Cov(Z|y_{1:m})$
and bounds the latter covariance by a constant, since $Z$ is supported on a bounded set (as the footnote in page 18 suggests).
However, it seems conceivable to me that if we do not impose the assumption s $X = Z + N_0$, we could instead
show directly that $\mathbb{E}[Cov(X|y_{1:m})]$ decreases as $O(1/m)$ if $X$ is bounded. The reason I believe this is true is that, by Proposition 2, we know that the posterior
of $X$ conditioned on $y_{1:m}$ converges in distribution to $X$ at a rate of $1/m$. Thus, the conditional variance of $X$ given $y_{1:m}$
should decrease as $m$ becomes larger. Another intuitive way to see this is that we could for example simply take the average
$\overline{y_m}$, which is $O(\sigma/m)$ close to $X$. Thus, we gain more and more information about $X$ as $m$ increases, which suggests
that $\mathbb{E}[Cov(X|y_{1:m})] = O(1/m)$ always. Have the authors considered this more general statement?

---

> ### Author Response · Authors · 2023-11-20
>
> Thank you for your thorough review and the suggestions for improving the paper.
>
> Regarding **Weaknesses**:
>
> We have expanded Sec. 4.2.1 on the important problem of estimating $g(y; \sigma)$, however due to space we moved the details to the appendix (Appendix F in the revised manuscript). We have also added more experiments to Appendix I using Langevin MCMC for estimating the smoothed score functions.  The question the reviewer has raised on the difficulty of estimating $g(y ; \sigma)$ is very important (we had a brief discussion of it in Sec. 6) and theoretical analysis for this aspect of our algorithm is a great research direction.
>
> Regarding **Questions**:
>
> - We added a new section to the appendix, Appendix F: Estimators for $\nabla \log p(y)$, giving the derivations for the estimators (F.1) (the same as (4.8) in our original submission) and (F.2) for the Langevin MCMC estimator. In that section, we also give the algorithm (Algorithm 3) for estimating $g(y; \sigma)$ using (overdamped) Langevin MCMC.
>
> - This is a great observation. Our main intuition here is that for any degree of smoothing sampling should become easier upon accumulation of noisy measurements since we're gathering more "evidence" on the clean sample, but this is too crude of an intuition perhaps. This observation by the reviewer is very interesting and perhaps there are some deeper results here related to the point below.
>
> - We did consider strengthening the theorem along the lines of the reviewer's comments, in particular whether $1/m$ rate still holds when $\tau=0$. Please note that in Theorem 1 (as opposed to Lemma 2) we are interested in the conditional Hessian itself, not the expected conditional Hessian. The main challenge here is that Theorem 1 is currently too restrictive (almost surely) and we think a great future direction is to see under what conditions the $1/m$ rate holds with *high probability*. This discussion is also related to our Remark 3 in the paper on the hardness of strenghtening  Theorem 1 regarding the monotonicity (4.7). We appreciate the reviewer sharing their thoughts on this aspect of the problem: although the analysis of the expectation of the Hessian is interesting in its own terms, it cannot be directly used to provide some guarantees on progressively more log-concave sampling.

---

### Official Review · Reviewer_gqez · 2023-11-01

**Soundness:** 3 good
**Presentation:** 4 excellent
**Contribution:** 3 good
**Rating:** 6
**Confidence:** 3

**Summary:**

The paper introduces an MCMC-type method that sequentially samples an unnormalized density with Gaussian smoothing. The critical theoretical contribution shows that with a proper choice of noise level and the sequential strategy, the conditional densities with compact support (or similar ones) become more and more log-concave "healthily." Through the process, the algorithm mainly keeps the means, given the properties of the Gaussian kernels. Internally, the algorithm employs an MCMC strategy (like Langevin) and methods for score function estimation.

**Strengths:**

The submission is a well-written research paper. The authors explained the concepts clearly and presented the critical theoretical contribution elegantly. The authors have done an excellent job of demonstrating the effectiveness of the proposed method and its potential applications.

The problem itself is also fundamental and worth studying. The authors start by finding that "all (measurements) at once" (AAO) is suboptimal and then present a "one (measurements) at once" (OAT) algorithm that has solid theoretical guarantees. In particular, Gaussian smoothing helps transform a (nearly) compact density towards log-concavity so that it's feasible to sample from it, and measurements accumulation (in a non-Markovian manner) makes the density more and more log-concave and well-conditioned. Combining both effects makes the proposed OAT strategy superior to AAO ones.

Moreover, the paper demonstrates the value of combining theory and experiments in a research project. The theoretical analysis provides a solid foundation, while the experiments validate the proposed method and help readers gain more insights and understand its accuracy. Combining theory and experiments leads to a more complete understanding of the OAT strategies and opens up new avenues for further research.

**Weaknesses:**

One weakness is in the result presentation: The paper does not provide a direct performance guarantee on Algorithm 1. I understand the inner-loop method is the object of study here. Still, it should be feasible to make some assumptions around its properties and derive a theorem that captures the algorithm's performance as $m$ and $n_t$ increase. The error metric could ideally be a Wasserstain-type distance for consistency with the experiment section.

The experiment evaluation partially supports and reflects the theoretical claims, but the settings could be more complex for a thorough evaluation. For example, it would be interesting to understand what magnitude the # sampling iterations must be for some rather complex distributions, e.g., $k$-mixture of Gaussians where $k$ increases. The densities used in Section 5 are well-designed to test specific hypotheses/claims. Yet, experiments should probably cover more than that, such as demonstrating the properties of the main algorithm(s), which can be critical for method adoption.

Minor points:
- May I know the purpose of "4.1 Example" as it seems to be just deriving various quantities for a concrete/standard Gaussian mixture? I don't see how it helps the readers better understand Theorem 1.
- In Section 5.2, is there an intuitive explanation for why "the optimal $\sigma$ here is in fact larger than the noise level needed to make $p(y_1)$ log-concave"?
- Would it make sense to add more description for Figure 4, panels (c) and (d)? Currently, it is not entirely clear what they are offering.

**Questions:**

Please take a look at the "Weaknesses," where the comments cover both theory presentation and experiments, as well as some additional questions (under "minor points").

One additional question:
- Is the actual code anonymously shared somewhere? It would be good to have it for result reproducibility and smooth adoption of the method.

---

> ### Author Response · Authors · 2023-11-20
>
> Thank you for your supportive review and the suggestions for improving the paper.
>
> Regarding **Weaknesses**:
>
> - That direct performance guarantee on Algorithm 1 is an important problem to tackle in understanding and extending the formalism developed in this work. We hope the reviewer reconsider this as a weakness if it has affected their score. The paper is already dense with theoretical results and we had limited space for supporting the theoretical formalism with experiments on demonstrating the effectiveness of smoothing and the log-concave sampling strategy which the reviewer has kindly highlighted.
>
> - The experiment on $k$-mixture of Gaussians as $k$ and $d$ increase is very interesting, but requires careful considerations in experimental design... We are currently working on extending our work to the Ising model on the Boolean hypercube {$-1,1$}$^d$, where, after smoothing, one can effectively think of as $k$-mixture of Gaussians with $k=2^d$ mixtures.
>
> Regarding minor points:
>
> - Thank you for highlighting this. The proof of Theorem 1 is somewhat involved and we found it useful to give an example where the reader can easily follow all the derivations. We also like Example 4.1 since it points to some stronger theoretical results than what we could prove in this paper, e.g., the upper bound in Theorem 1 becomes tight in the example we considered.
>
> - Our intuitive explanation is that larger $\sigma$ leads to better condition numbers which seems to help beyond the log-concavity threshold. This is however dependent on the MCMC$_\sigma$ used in Algorithm 1.
>
> - We have added more descriptions to the caption for Figure 4, panels (c) and (d), which illustrates that the OAT scheme samples the subdominant mode well, whereas very few samples reach this mode when $\sigma=0$.
>
> Regarding **Questions**: We have shared the code for this submission [at this anonymous link](https://anonymous.4open.science/r/solaris-1FC4) and will publicly release the code upon acceptance of the paper.

---

### Official Review · Reviewer_9mSy · 2023-11-05

**Soundness:** 4 excellent
**Presentation:** 3 good
**Contribution:** 4 excellent
**Rating:** 8
**Confidence:** 3

**Summary:**

This paper proposes a framework of sampling from unnormalized densities by transforming this problem to more amenable log-concave sampling.

**Strengths:**

**Originality**: The paper is an extension of the paper by Saremi & Srivastava. The proposed framework is very different from what I could find in the literature.

**Quality**: Once familiar with the literature (see weaknesses), I found the framework really elegant and simple. I learnt a lot from reading this paper.

**Clarity**: The paper is well-written (again see the caveat in the weaknesses section) for someone already familiar with this literature. The calculations are thorough without being burdensome. The explanations are crisp without being terse.

**Significance**: The final algorithm proposed is simple and should be easily adopted by the community. More importantly, I found the paper useful in learning how to think about sampling, and I believe it will be useful for the rest of the community also.

**Weaknesses:**

The paper could have done a better job at exposition of not so well-known notions like walk-jump sampling. The current exposition is good enough for someone intimately familiar with the literature, but for others, like me, it takes some reading of cited papers to not see ideas as being "pulled out of a hat".

**Questions:**

1. How to handle the case when the desired probability measure does not have a density?
2. It would have been insightful to discuss adversarial cases where the proposed sampling technique fails.

---

> ### Author Response · Authors · 2023-11-20
>
> Thank you for your review. We are delighted to see that you enjoyed our paper.
>
> We appreciate the time you took on the literature on empirical Bayes and walk-jump sampling. We will take the reviewer's comment into consideration in follow-up work.
>
> Regarding **Questions**:
>
> 1. We are not sure if the reviewer is referring to discrete distributions here. We are currenly working on extending our work to sampling from the Ising model where the random variable takes values in the Boolean hypercube \{$-1,1$\}$^d$. In our formalism, after smoothing sampling becomes equivalent to sampling from a (smoothed) k-mixture of Gaussians with $k=2^d$ mixtures.
>
> 2. Not adversarial per se, but we have included experiments in the paper that point to the limitations of the framework, mainly regarding the problem of score estimation. In short, the performance of the algorithm suffers if we the estimator of $g(y; \sigma)$ (used in Algorithm 2, in turn in Algorithm 1) is poor. This was briefly discussed in the original submission in Sec. 6. Please also see Appendix F and I in the updated manuscript.

---

> > ### Comment · Reviewer_9mSy · 2023-11-21
> > **Reply to authors**
> >
> > Thank you for the reply. I will keep my score and think that this paper is a valuable contribution.

---

### Official Review · Reviewer_ZJfB · 2023-11-07

**Soundness:** 3 good
**Presentation:** 4 excellent
**Contribution:** 3 good
**Rating:** 6
**Confidence:** 4

**Summary:**

This paper presents a framework for sampling from an unnormalized density $p\propto{\rm e}^{-f}$ on $\mathbb{R}^d$ by using a Gaussian smoothing scheme. It constructs $X\sim p$, $Y_i|X\sim_{\rm i.i.d.}\mathcal{N}(X,\sigma^2I)$ ($i\in[m]$) for some suitable $\sigma$ and $m$. By sequentially (one at a time) sampling the joint distribution of $Y_{1:m}$, the conditional distributions will become "more log-concave", and finally it outputs $\mathbb{E}[X|Y_{1:m}]$ as an approximate sample from $p$, which can be expressed in a function of $\bar{Y}_{1:m}$. The paper demonstrates the advantages of the sequential sampling scheme through some theoretical and empirical examples.

**Strengths:**

Overall, the paper introduces a novel theoretical framework that differs from existing MCMC methods or sequential inference models. The paper is well-written, with clear notations, convincing examples, and mathematically rigorous proofs. The structure is well-organized: the authors first introduce an example of an anisotropic Gaussian target distribution, which leads to the important observation that OAT is easier than AAO in terms of the condition number. Then, they present the theory for general distributions (once log-concave, always log-concave), which helps the reader understand the motivation.

**Weaknesses:**

Score estimation is essential for the algorithm's efficiency. However, most of the experiments in the paper use the analytic score function, which is not realistic in practice. Moreover, the score has to be estimated at each step of ${\rm MCMC}_\sigma$, which will significantly increase the algorithm's complexity.

The paper only compares the performance of accurate and approximate scores in appendix H, and the result is not very satisfactory. The estimated score performs well when $d=2$, but it deteriorates when $d=8$. These results suggest that the score estimate's precision could be much worse in high dimensions, which I believe is the paper's main weakness.

In 4.2.1, the proposed method for score estimation is based on importance sampling (which transforms the expectation w.r.t. the probability density $\nu\propto\exp(-f-\frac{1}{2\sigma^2}||\cdot-y||^2)$ to $\mu=\mathcal{N}(y,\sigma^2I)$). However, (Chatterjee and Diaconis, 2018) has shown that the sample size needed for accurate estimation by importance sampling is usually $\exp({\rm KL}(\nu||\mu))$ for general target functions. For this reason, (Huang et al, 2023, section 3) use Langevin Monte Carlo to sample from $\nu$ and combine it with importance sampling in a similar task of score estimation. The experiments in your paper indicate that the score estimator also suffers from high variance. The authors should experiment with different score estimators, instead of saying that "studying the covariance of the plug-in estimator and devising better estimators is beyond the scope of this paper".

References:

(Chatterjee and Diaconis, 2018) Sourav Chatterjee, Persi Diaconis. The sample size required in importance sampling. Ann. Appl. Probab. 28(2): 1099-1135 (April 2018). DOI: 10.1214/17-AAP1326.

(Huang et al., 2023) Xunpeng Huang, Hanze Dong, Yifan Hao, Yian Ma, Tong Zhang. Reverse Diffusion Monte Carlo. ArXiv preprint arXiv:2307.02037.

**Questions:**

1. The algorithm outputs a *biased* sample of $p_X$, but the bias $W_2^2(p_X,\hat{p}_{m^{-1/2}\sigma})$ can be made arbitrarily small by choosing $\sigma$ and $m$ such that $\frac{\sigma^2d}{m}$ is small enough. How can we choose these parameters in a principled way? In the Gaussian example (proposition 4), we can reduce $\kappa_1$ by increasing $\sigma$, and in theorem 1, we need a large $\sigma$ to make $p(y_1)$ strongly log-concave. However, this also increases the bias, so we need a larger $m$ to compensate that. How can we balance this trade-off between complexity and accuracy?

2. In the experiment, how do you choose the projection direction $\theta$ for estimating the Wasserstein-2 distance? A more fair comparison would be to use $\theta$ following the uniform distribution on the Euclidean unit ball (i.e., the *sliced* Wasserstein-2 distance). Efficient approximation algorithms are available, for example, in (Nadjahi et al., 2021).

Minor corrections:

1. In A.1, $-2\sigma^2\log p(y_{1:m}|x)=\sum_{t=1}^{m}||y_t-x||^2+{\rm cst}$.

2. In remark 3, for (4.7) to hold, ${\rm cov}(Z|y_{1:m})\preceq{\rm cov}(Z|y_{1:m-1})$ almost surely; ... is very large, so that ${\rm cov}(Z|y_{1:m-1})\approx0$.

3. The exponential family in the third footnote should be $p(x|\nu)=\exp(-f(x)+\frac{1}{\sigma^2}(x-x_0)^\top(y-x_0)-\frac{\nu}{2\sigma^2}(||x-x_0||^2+||y-x_0||^2)-a(\nu))$.

4. In the third paragraph of introduction and the second paragraph of section 2, $Y_t=X+N_t$, $t\in[m]$. $N_t$ are i.i.d. $\mathcal{N}(0,\sigma^2I)$ random variables that are also independent of $X$. In theorem 1, $X=Z+N_0$, $Z$ and $N_0$ are independent.

References:

(Nadjahi et al., 2021) Kimia Nadjahi, Alain Durmus, Pierre Jacob, Roland Badeau, Umut Şimşekli. Fast Approximation of the Sliced-Wasserstein Distance Using Concentration of Random Projections. In Advances in Neural Information Processing Systems, 2021.

---

> ### Author Response · Authors · 2023-11-20
>
> Thank you for your thorough review and the suggestions for improving the paper. We updated our experiments to incorporate the comments, and we hope the reviewer would revise their score in light of the changes made to the manuscript.
>
> Regarding **Weaknesses**:
>
> We agree with the reviewer that estimating the score function using Langevin MCMC is better than using the estimator we experimented with for the results in the appendix. Following the submission of the paper we did conduct experiments where we used Langevin MCMC for estimating the smoothed score functions. We modified Sec. 4.2.1 and added a new section to the appendix (Appendix F in the revised manuscript). We have also updated the results in Appendix I with the new experiments.
>
> Thank you for pointing out the reference (Chatterjee and Diaconis, 2018). This problem is also related to the literature on "Entropy SGD", e.g., (Chaudhari et al., 2017), where the estimating smoothed score functions (although it's not framed like that) is used in the context of optimizing neural network parameters. (Chatterjee and Diaconis, 2018) and (Chaudhari et al., 2017) are now cited -- please see Appendix F. (Huang et al, 2023) is a concurrent work, which we were not aware of. We would be happy to cite it in the camera-ready after a careful reading of the paper.
>
>
> We consider the new experiments added to the appendix (Fig. 7) on running Algorithm 1 using score estimation with Langevin MCMC as preliminary as we did not conduct exhaustive tuning of the underdamped Langevin MCMC used for the score estimation experiments.
>
> Regarding **Questions**:
>
> 1. The reviewer is correct in pointing out the fundamental trade-off in our formalism: larger $\sigma$ makes sampling easier but then we require more measurements to get close to the target distribution. We currently do not have a bound on convergence of our algorithm taking everything into consideration. Another layer of complexity is the computational complexity of estimating $g(y; \sigma)$. This is discussed around two main limitations of our work in Sec. 6. This is an important future direction for further understanding the formalism we have developed in this work.
>
> 2. We agree with the reviewer that sliced 2-Wasserstein distance with random projections is more suitable than the 1D marginal 2-Wasserstein distance reported in the paper (for elliptical Gaussian we used the "difficult" direction). In the revised manuscript uploaded here, the metric is updated to the sliced 2-Wasserstein distance. We did not use the efficient approximation framework by Nadjahi et al. 2021, but implemented the 2-Wasserstein distance using 1000 random projections (Eq. 9 in the reference).
>
> Thank you for your minor corrections! They have been incorporated in the manuscript.
>
> ----
> Pratik Chaudhari, Anna Choromanska, Stefano Soatto, Yann LeCun, Carlo Baldassi, Christian Borgs, Jennifer Chayes, Levent Sagun, and Riccardo Zecchina. Entropy-SGD: Biasing gradient descent into wide valleys. In *International Conference on Learning Representations*, 2017.

---

> > ### Comment · Reviewer_ZJfB · 2023-11-22
> >
> > Dear authors,
> >
> > I sincerely appreciate your efforts in addressing my concerns and updating the paper. The connection between score estimation and entropy SGD is indeed an intriguing aspect, and I acknowledge its importance for future research.
> >
> > In the revised Appendix I, the inclusion of the Langevin estimator of the score alongside the comparison with the importance sampling estimator is valuable. However, once again, neither of the methods attains a performance comparable to the analytic score, especially in high dimension ($d=8$). In my view, this limitation represents a significant drawback of the proposed framework and similar methods such as the reverse diffusion Monte Carlo (Huang et al., 2023) which requires efficient score estimation of a Gaussian-convoluted density without incorporating learning components.
> >
> > Despite my genuine appreciation for the novelty of the framework and the overall quality of the paper, the observed limitations in practical performance leave me unconvinced regarding the algorithm's efficiency. Regrettably, I find myself unable to assign a higher score to this paper. I believe that addressing the challenges associated with score estimation could significantly enhance the practical applicability of your framework.

---

### Author Response · Authors · 2023-11-20

We thank the reviewers for their enthusiastic and supportive reviews of our paper, and the comments and questions that have led us to improve our paper. We have carefully considered the reviewers’ feedback, prepared a point-by-point response to each reviewer, and updated the manuscript. The main part of the paper (theoretical results) has remained unchanged, but we now use the sliced 2-Wasserstein metric, as suggested by Reviewer ZJfB, in our experiments. We expanded Sec. 4.2.1 on the problem of estimating smoothed score functions, but due to space the new materials are moved to the appendix (Appendix F). We also conducted a more exhaustive hyperparameter search following the submission of the paper and Fig. 3 is updated as such.

---

### Meta-Review · Area_Chair_Ztx3 · 2023-12-04

**Metareview:**

This paper focuses on constructing an algorithm that efficiently generates samples from an unnormalized density, beyond the standard log concave situation. This is enabled by an innovative non-Markovian chain of log-concave Markov Chains. All reviewers and I agree that the ideas are interesting and, despite that there are still some concerns from the reviewers (e.g., estimation of the score function), I recommend acceptance.

I request the authors to revise the paper where appropriate, based on the discussion with the reviewers.

**Justification For Why Not Higher Score:**

I agree with Reviewers ZJfB & gqez's concerns. The empirical demonstration could be stronger.

**Justification For Why Not Lower Score:**

The idea is innovative and the theoretical demonstration is solid.

---

### Decision · Program_Chairs · 2024-01-16

Accept (poster)